# A polymer controlled nucleation route towards the generalized growth of organic-inorganic perovskite single crystals

Lin Ma [1,2], Zhengguang Yan[1,2 ✉], Xiaoyuan Zhou [3 ✉], Yiqun Pi[1,2], Yiping Du[1,2], Jie Huang[1,2], Kaiwen Wang[1,2], Ke Wu[4], Chunqiang Zhuang[1,2] & Xiaodong Han [1,2 ✉]

Recently, there are significant progresses in the growth of organic-inorganic lead halide perovskite single crystals, however, due to their susceptible nucleation and growth mechanisms and solvent requirements, the efficient and generalized growth for these single crystals is still challenging. Here we report the work towards this target with a polymer-controlled nucleation process for the highly efficient growth of large-size high-quality simple ternary, mixed-cations and mixed-halide perovskite single crystals. Among them, the carrier lifetime of $FAPbBr_3$ single crystals is largely improved to 10199 ns. Mixed $MA/FAPbBr_3$ single crystals are synthesized. The crucial point in this process is suggested to be an appropriate coordinative interaction between polymer oxygen groups and $Pb^{2+}$, greatly decreasing the nuclei concentrations by as much as 4 orders of magnitudes. This polymer-controlled route would help optimizing the solution-based OIHPs crystal growth and promoting applications of perovskite single crystals.

[1] Institute of Microstructure and Property of Advanced Materials, Faculty of Materials and Manufacturing, Beijing University of Technology, Beijing, China. [2] Beijing Key Laboratory of Microstructure and Properties of Solids, Beijing University of Technology, Beijing, China. [3] College of Physics and Institute of Advanced Interdisciplinary Studies, Chongqing University, Chongqing, China. [4] Beijing National Laboratory for Molecular Sciences, State Key Laboratory of Rare Earth Materials Chemistry and Applications, PKU-HKU Joint Laboratory in Rare Earth Materials and Bioinorganic Chemistry, College of Chemistry and Molecular Engineering, Peking University, Beijing, China. ✉email: yanzg@bjut.edu.cn; xiaoyuan2013@cqu.edu.cn; xdhan@bjut.edu.cn

Organic–inorganic halide perovskites (OIHPs) express impressive performance in solar cells[1–4], light-emitting diodes[5–7], lasers[8,9], ultraviolet-to-infrared photodetectors[10,11], x-ray[12–14], γ-ray detectors[15–17], and x-ray scintillators[18,19]. The OIHPs single crystals show great promise in these applications, for instance, a sensitive x-ray detector made of MAPbBr$_3$ perovskite single crystals exhibited the most sensitive detectable x-ray dose rate of 0.5 μGy$_{air}$ s$^{-1}$ with a sensitivity of 80 μGy$^{-1}_{air}$ cm$^{-2}$, which was four times higher than α-Se x-ray detectors[12]. A well-defined $^{137}$Cs energy spectrum was obtained by using a CH$_3$NH$_3$PbBr$_{2.94}$Cl$_{0.06}$ single crystal under a small electric field of 1.8 V mm$^{-1}$ at room temperature[16]. The CsPbBr$_3$ nanocrystal scintillator for nondestructive x-ray imaging was highly sensitive with a detection limit of 13 nGy s$^{-1}$, which was about 400 times lower than the typical medical-imaging doses[18].

The growth of large-size high-quality perovskite single crystals has drawn increasing interest. Many significant progresses have been made in the growth of OIHPs single crystals. For example, MAPbI$_3$ single crystal with size up to 71 mm with aid of a seed-induced heterogeneous nucleation was fabricated[20] and a low-temperature-gradient crystallization method for high-quality MAPbBr$_3$ single crystals was developed[21]. However, due to their versatile nucleation and growth mechanisms and solvent requirements, an efficient and generalized growth for OIHPs single crystals is still challenging. The classical solution crystal growth methods including the solution temperature-lowering (STL) route, the anti-solvent vapor-assisted crystallization (AVC) method, and slow evaporation (SE) method have been employed for the synthesis of OIHPs single crystals[22]. Although the STL method is considered a convenient and effective approach for the growth of large-size high-quality single crystals, this method is time consuming to obtain a 1-cm-sized single crystal[23,24]. The AVC method avoids the temperature-dependent phase transitions in crystals while it is challenging to obtain large size single crystals[25]. The inverse temperature crystallization (ITC) method based on the inverse temperature dependence of solubility is developed to prepare OIHPs single crystals with high growth rate driven by a high supersaturation[20,21,26,27]. While, the rapid growth induced by the high supersaturation is usually accompanied with rapid formation of large number of nuclei, which would lead to numbers of small single crystals. High supersaturation could also weaken the stability of growth solution so that impurity phase and defect structures could appear. By supplying a seed crystal, the spontaneous nucleation could not be completely avoided, and the impurity phase and defect structures may appear[28].

In this work, we demonstrate that the polymer ligands, such as polyethylene glycol (PEG), polypropylene glycol (PPG), polyacrylic acid (PAA), and polyvinyl alcohol (PVA), all containing oxygen groups, can significantly enhance the stability of OIHPs growth solution. With the help of these polymers, the nucleation process can be well controlled and large-size high-quality OIHPs single crystals are obtained at high growth rate. We ascribe this polymer-controlled (PC) nucleation process to the coordinative interaction between the oxygen groups and Pb$^{2+}$ ions. This PC nucleation approach is potentially applicable for the other OIHPs single crystal growth.

## Results and discussions

**Single crystal growth and characterizations**. Figure 1 shows the images of several series of OIHPs single crystals synthesized through the PC route. The MAPbX$_3$ (X = I, Br, Cl), FAPbX$_3$ (X = I, Br), and CsPbBr$_3$ single crystals are shown in Fig. 1a, and the mixed-halide and mixed-organic cation OIHPs single crystals (MAPbI$_x$Br$_{3-x}$, MAPbBr$_x$Cl$_{3-x}$, MA$_y$FA$_{1-y}$PbBr$_3$, and MA$_y$FA$_{1-y}$PbI$_3$) are shown in Fig. 1b–d, respectively. The sizes of these

OIHPs single crystals are shown in Supplementary Table 1. It can be noticed that the Br-based single crystals show good transparency. These transparent OIHPs crystals may provide potential applications in optical devices. The synthesis of three series OIHPs of simple, mixed-halide and mixed-organic cation single crystals demonstrates the capability and flexibility of this PC route.

The crystal growth process begins with OIHPs powders, which are synthesized by a water bath method and are dissolved subsequently in the organic solvents to obtain the precursor solutions, then the single crystals are grown assisted by the polymers. The experimental conditions, including polymer, solvent, temperature and so on are shown in Supplementary Figs. 1–3 and Supplementary Tables 2, 3.

The synthesized OIHPs single crystals by the PC route are with high quality. The Supplementary Figs. 4–8 show the x-ray diffraction patterns, UV–vis–NIR absorption spectra, photoluminescence (PL) emission spectra, and x-ray rocking curves of some example crystals. From the x-ray rocking curves, the FWHM values of the FAPbI$_3$, MAPbI$_3$, FAPbBr$_3$ and MAPbBr$_3$ single crystals are measured as 0.118°, 0.236°, 0.174°, and 0.127°, respectively, which indicate that these single crystals are of high crystallinity and without the presence of impurity crystals.

The carrier mobility ($μ$) and carrier lifetime ($τ$) are measured and the carrier diffusion length ($L_D$) is calculated for the FAPbBr$_3$, FAPbI$_3$, MAPbI$_3$, and MAPbBr$_3$ single crystals as shown in Fig. 2, Table 1, and Supplementary Figs. 9, 10. In the transient state PL decay curves, the perovskite single crystals exhibit a superposition of fast and slow dynamics. As a comparison, some literature data[21,25–30] are included in Table 1. Based on the previous researches, the highest carrier lifetimes (slow component) of FAPbBr$_3$ and FAPbI$_3$ were ~2272 and ~839 ns, respectively[27,29]. In the current research, the carrier lifetimes of FAPbBr$_3$ and FAPbI$_3$ single crystals are largely promoted to be 10199 and 1393 ns, as shown in Fig. 2a and b, respectively. The MA$_y$FA$_{1-y}$PbBr$_3$ single crystals also show exceptional physical properties. As shown in Supplementary Fig. 11, the carrier lifetimes of MA$_{0.16}$FA$_{0.84}$PbBr$_3$ and MA$_{0.33}$FA$_{0.67}$PbBr$_3$ reach 8712 and 8420 ns, respectively. Compared with the literatures, some perovskite single crystals prepared by the PC route have much higher carrier lifetime as well as longer carrier diffusion length as shown in Fig. 2c and Table 1. Figure 2d shows the correlations between carrier lifetime and average single crystal growth rates (in volume) based on our work and literatures, indicating the high growth efficiency and the crystals' high quality of the PC method. The detailed information of the growth rate and carrier lifetime is shown in Supplementary Fig. 12 and Supplementary Tables 4, 5. These results indicate that the PC method provides a route for high-quality OIHPs single crystals growth.

**The effect of polymers on crystallization process**. To clarify the roles of the polymers and the PC synthetic mechanisms, the effect of polymers on the precursor solutions and crystallization process are systematically studied. The FAPbI$_3$/GBL precursor solutions with different concentration of PPG-3000 are measured by dynamic light scattering (DLS)[31,32], zeta potential, Raman, and UV–Vis absorption spectroscopy. As shown in Fig. 3a, the average colloid size of the FAPbI$_3$/GBL precursor solution measured is about 0.8 nm and the addition of PPG-3000 increases the colloids size up to 1.4 nm. The PPG-3000 can also increase the colloid size in FAPbBr$_3$, MAPbI$_3$, and MAPbBr$_3$ precursor solutions (inset Fig. 3a). Figure 3b further shows that the zeta potential ($ζ$) values are −15.3 and −22.0 mV for the FAPbI$_3$/GBL solution ($C_{FAPbI3} = 0.094$ g/mL) and FAPbI$_3$/(GBL + PPG-3000) ($C_{FAPbI3} = 0.094$ g/mL, $C_{PPG-3000} = 0.006$ g/mL) solution, respectively. The colloid size and the zeta potential increase upon the

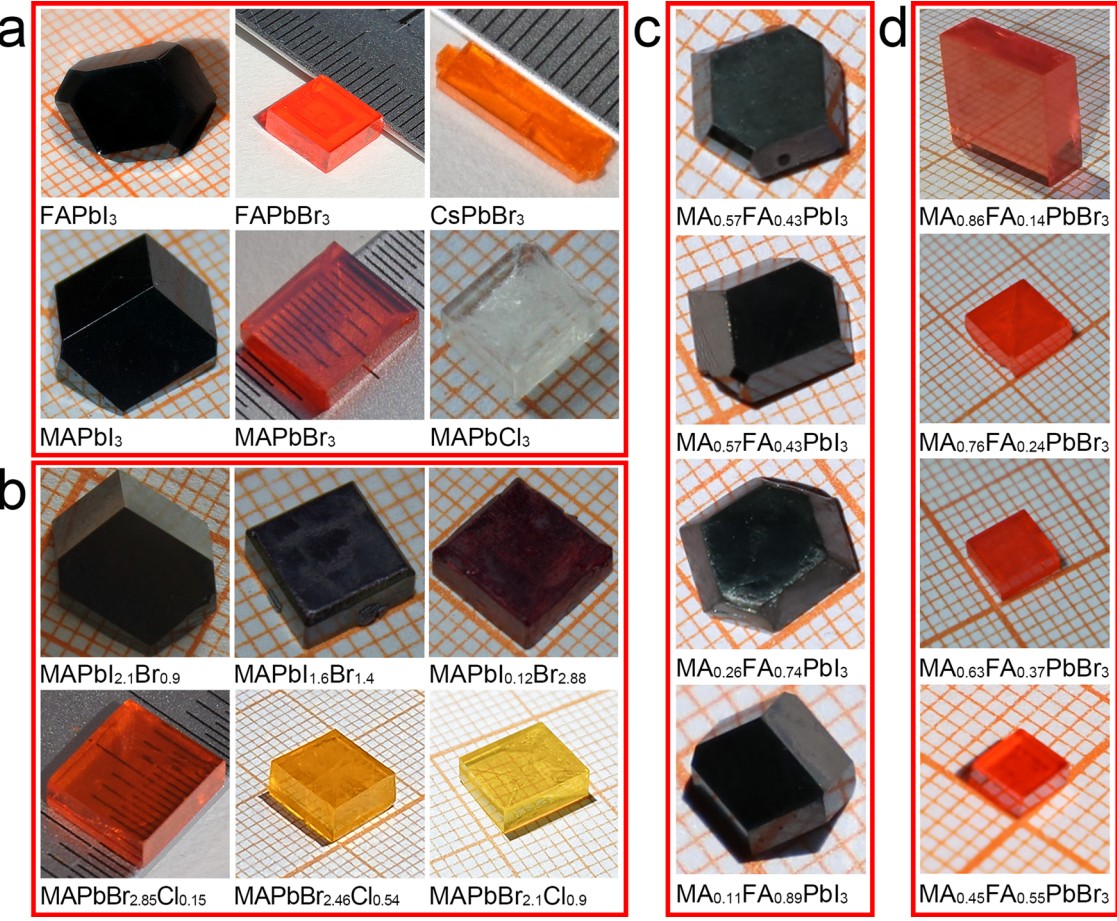

**Fig. 1 Photos of OIHPs single crystals. a** FAPbX$_3$ (X = I, Br), CsPbBr$_3$, and MAPbX$_3$ (X = I, Br, Cl) single crystals. **b** A series of mixed-halide MAPbI$_x$Br$_{3-x}$ and MAPbBr$_x$Cl$_{3-x}$ single crystals (0 ≤ x ≤ 3). **c**, **d** The mixed-organic cation MA$_y$FA$_{1-y}$PbX$_3$ (X = I, Br) single crystals (0 ≤ y ≤ 1).

introduction of PPG-3000 and this indicates that the iodo-plumbates (PbI$_n^{(n-2)-}$) colloids are attached to PPG-3000 molecules and result in the decrease of free PbI$_n^{(n-2)-}$ in FAPbI$_3$/GBL solution.

The Raman investigation further demonstrates that the peaks for Pb–I bonds decrease with PPG-3000 addition. As presented in Fig. 3c, there are two peaks around 100 and 122 cm$^{-1}$ ascribed to the Pb–I bonds[33]. Due to the presence of GBL, a blue shift of these two bands occurs compared with the literature. With the increase of PPG-3000 concentration, the intensity of the peak corresponding to the Pb–I bonds decreases. When the PPG-3000 concentration reaches 0.2 g/mL, the Raman peak intensity is reduced by over 50% in contrast to that of FAPbI$_3$/GBL precursor solution (0.75 g/mL). The results indicate that the PbI$_n^{(n-2)-}$ concentration decreases due to the attachment of PPG-3000. The UV–Vis absorption spectra further confirm that when different concentrations of PPG-3000 are added into the precursor solution, the absorption of PbI$_n^{(n-2)-}$ decreases. The iodoplum-bates (PbI$_n^{(n-2)-}$) with different n values have absorption bands in different wavelengths, such as 276, 320, and 370 nm are corresponding to PbI$_2$, PbI$_3^-$, and PbI$_4^{2-}$, respectively[34]. After adding PPG-3000 to the FAPbI$_3$/GBL solution (0.2 mM), the absorption bands around 276 nm (PbI$_2$), 322 nm (PbI$_3^-$), and 370 nm (PbI$_4^{2-}$) greatly reduced and even disappeared, while peaks around 294 nm (I$_3^-$) and 365 nm (I$_3^-$) appear as shown in Fig. 3d. The changes in spectra that confirm the polymer replace the I$^-$ in the polyhalide coordination.

Finally, the coordination interaction of oxygen groups of PPG-3000 with Pb$^{2+}$ ions are verified by Raman and Fourier-transformed infrared spectroscopy (FTIR) (Supplementary Figs. 13, 14 and Supplementary Table 6). In the Raman spectra of α-FAPbI$_3$/PPG-3000 powder, a new peak corresponding to the Pb–O bond is observed. The FTIR spectra of α-FAPbI$_3$/PPG-3000 solutions reveal the shift of C–O–C peaks towards lower frequency which derives from the formation Pb–O coordination bonds.

For the OIHP crystallization process, we study the crystal sizes at different growth stages to realize the effect of polymers by transmission electron microscopy (TEM) with the liquid-cell TEM method[35–37]. For comparison, the conventional dry-up sample preparation method is also applied to monitor the crystal growth process. Figure 4a presents that the FAPbI$_3$ initial nanocrystals are a few hundred nanometers in size in the FAPbI$_3$/(GBL + PPG-3000) growth solution measured using the liquid-cell TEM method. As a comparison and shown in Fig. 4b, the FAPbI$_3$/(GBL + PPG-3000) growth solution without dilution is dripped onto TEM grids and dried for TEM observation. The initial nanocrystals can be identified as α-FAPbI$_3$ nanocrystals with similar sizes which confirm the results from liquid-cell TEM. Without PPG-3000, as shown in Fig. 4c, the initial nanocrystals in the FAPbI$_3$/GBL growth solutions, are only a few nanometers in size measured on the direct dripping samples. Summarized in Fig. 4d, the addition of PPG-3000 greatly changes the size of initial nanocrystals in the growth solution, in which, the average initial nanocrystals sizes of α-FAPbI$_3$ increase from ~3.3 to ~197.4 nm. Meanwhile, the nanocrystal concentration decreased from ~1000/μm$^2$ without PPG to ~1/μm$^2$ with PPG-3000 by TEM observation. This variation results from the decreased nucleation concentration upon the addition of PPG-

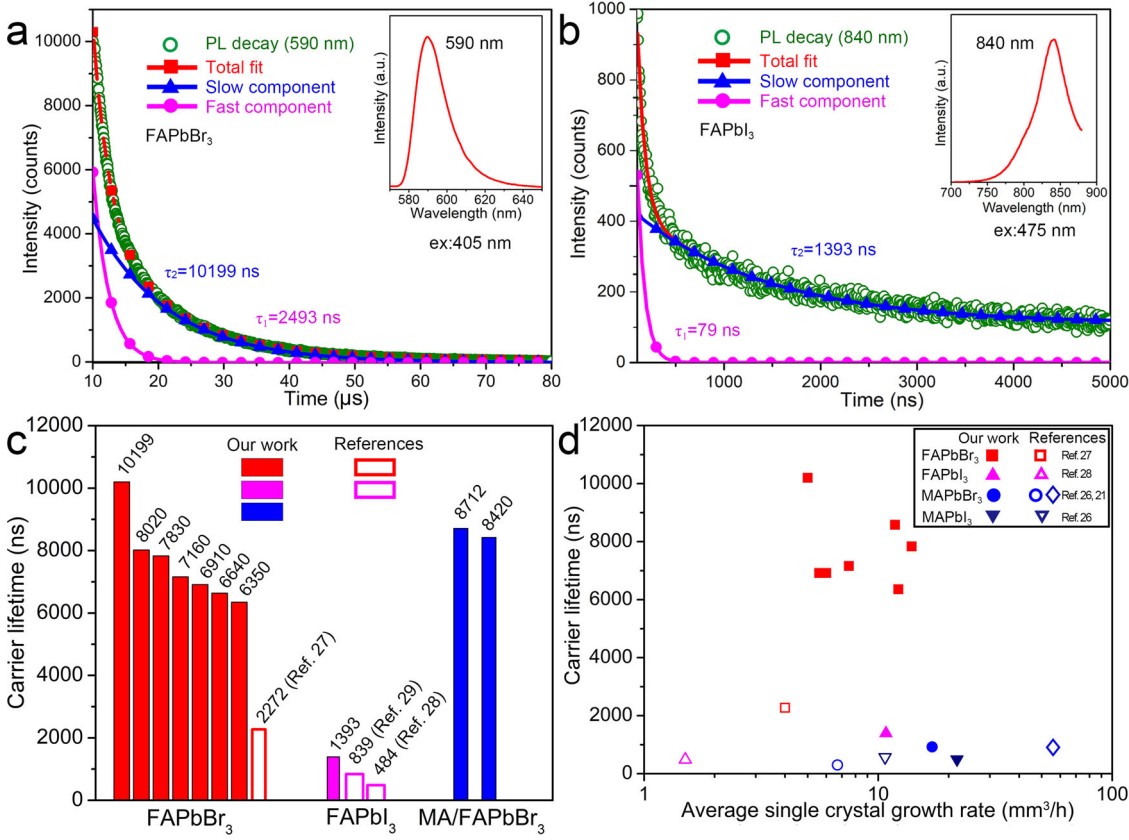

**Fig. 2 Carrier lifetime measurements.** Steady and transient state photoluminescence (PL) spectra are measured using a 405 and 475 nm excitation wavelengths, respectively. PL decay curves of **a** the FAPbBr$_3$ single crystal and **b** the FAPbI$_3$ single crystal. **c** Comparison of carrier lifetimes of OIHPs single crystals (seven FAPbBr$_3$ samples are tested)[27–29]. **d** The correlations between carrier lifetimes and average single crystal growth rates (in volume) based on our work and literatures[21,26–28].

**Table 1 Some typical physical properties of the OIHPs single crystals via the PC approach.**

| Refs. | OIHPs | $n_t$ (cm$^{-3}$) | $\mu$ (cm$^2$ V$^{-1}$ s$^{-1}$) | $\tau_1$ (ns) | $\tau_2$ (ns) | Worst-case $L_D$ (μm) | Best-case $L_D$ (μm) |
|---|---|---|---|---|---|---|---|
| Present work | FAPbBr$_3$ | $2.4 \times 10^{10}$ | 36 | 2493 | 10199 | 15.2 | 30.8 |
| | FAPbI$_3$ | $1.4 \times 10^{10}$ | 41 | 79 | 1393 | 2.9 | 12.1 |
| | MAPbBr$_3$ | $6.6 \times 10^9$ | 39 | 122 | 923 | 3.6 | 9.7 |
| | MAPbI$_3$ | $3.0 \times 10^{10}$ | 68 | 94 | 493 | 4.0 | 9.4 |
| 21 | MAPbBr$_3$ | $6.7 \times 10^9$ | 83.9 | 132 | 897 | 5.3 | 13.8 |
| 25 | MAPbI$_3$ | $3.3 \times 10^{10}$ | 2.5 | 22 | 1032 | 2 | 8 |
| | MAPbBr$_3$ | $5.8 \times 10^9$ | 38 | 41 | 357 | 3 | 17 |
| 26 | MAPbI$_3$ | $1.4 \times 10^{10}$ | 67 | 18 | 570 | 1.8 | 10.0 |
| | MAPbBr$_3$ | $3.0 \times 10^{10}$ | 24 | 28 | 300 | 1.3 | 4.3 |
| 27 | FAPbI$_3$ | $1.1 \times 10^{10}$ | 35 | – | – | 1.7 | 6.6 |
| | FAPbBr$_3$ | $9.6 \times 10^9$ | 62 | 687 | 2272 | 10.5 | 19.0 |
| 28 | FAPbI$_3$ | $6.2 \times 10^{11}$ | 4.4 | 32 | 484 | 0.5 | 2.2 |
| 29 | FAPbI$_3$ | – | – | 91 | 839 | – | – |
| | MAPbI$_3$ | – | – | 7 | 146 | – | – |
| 30 | MAPbBr$_3$ | $4.4 \times 10^9$ | – | – | 997 | – | – |

3000 which is the origin of the subsequent large size OIHPs crystals growth. Figure 4e illustrates PPG-3000 increases the size of APbX$_3$ (A = MA, FA; X = I, Br) single crystals with decreasing the numbers of single crystals.

To further confirm the initial nanocrystal sizes, the APbX$_3$ (A = MA, FA; X = I, Br) growth solution with or without PPG-3000 are characterized by DLS (Supplementary Fig. 15). Combined with the final OIHPs crystal sizes shown in Fig. 4e, a correlation of final single crystal sizes and the initial nanocrystal sizes is shown in Fig. 4f. Taking FAPbI$_3$ as an example, the initial

nanocrystal size increased two orders of magnitude and the corresponding single crystal size increased over five times compared to the growth solution without polymer. Therefore, The PPG-3000 decreases the nucleation concentration and consequently increases the initial nanocrystal size, which is critical for the subsequent large size OIHPs single crystal growth.

An overview of the growth process kinetics is further given by a real-time electrical conductivity measurement of the growth solution, which provides the information of reacting ions in the non-aqueous solution[38]. An in situ electrochemical impedance

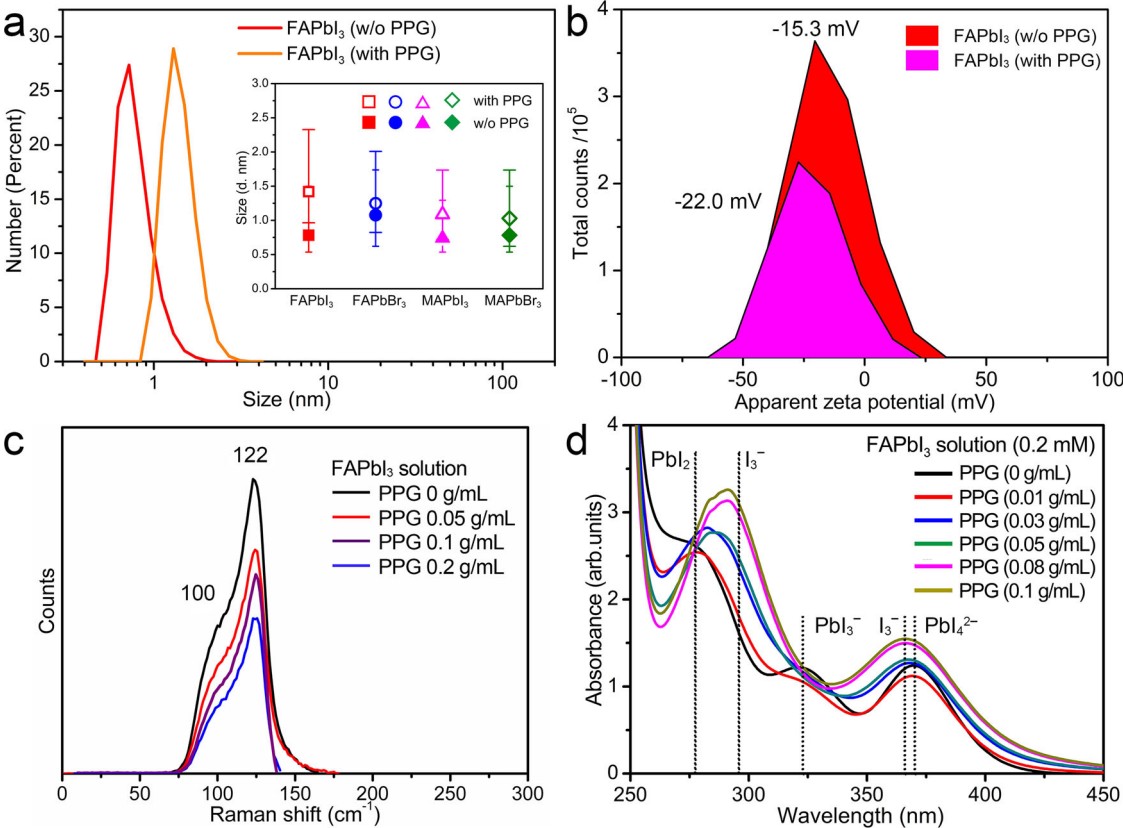

**Fig. 3 The effects of PPG-3000 addition on the growth solutions. a** The size distribution of colloid in precursor solutions with or without PPG-3000, measured by dynamic light scattering. The error bars of inset (**a**) indicate the colloid size range, note that the solution of PPG-3000/GBL without OIHPs does not present signals. **b** The zeta potential ($\zeta$) of the colloidal particles in the FAPbI$_3$ ($C_{FAPbI3} = 0.094$ g/mL) and FAPbI$_3$/(GBL + PPG-3000) solutions. ($C_{FAPbI3} = 0.094$ g/mL, $C_{PPG-3000} = 0.006$ g/mL). **c** Raman spectra of FAPbI$_3$ GBL solution (0.75 g/mL) with different concentrations of PPG-3000. **d** The UV–Vis spectra of FAPbI$_3$ solution ($C_{FAPbI3} = 0.2$ mM) with different concentrations of PPG-3000.

spectroscopy (EIS) method is used for probing the kinetics of OIHPs crystallization. The early crystallization process of the OIHPs crystals and the final single crystal growth with and without polymer can be evaluated correspondingly. The FAPbI$_3$/(GBL + PPG-3000), FAPbI$_3$/GBL, and FAPbI$_3$/(GBL + pentadecane) growth solutions are tested by EIS, respectively (Supplementary Fig. 16). Based on the equivalent circuit model, the $R_s$ (ohmic resistance of solution) can be calculated, which corresponds to the solution impedance moduli at the high frequency end ($|Z|$) (Supplementary Fig. 17). Meanwhile, the solution concentration as a function of $R_s$ at growth temperature is obtained (Supplementary Fig. 18) and the real-time concentrations of the growth solution is calculated according to the working curve. The real time solute consumption curves of FAPbI$_3$/(GBL + PPG) and FAPbI$_3$/GBL are shown in Fig. 5. The insets in Fig. 5 are the corresponding optical images of growth solutions with growth crystals at final stage.

In all three curves shown in Fig. 5, an apparent accelerated solute consumption process can be observed. We define the point marking the starting of increasing solute consumption rate as a point A and the point that the solute consumption rate drops to a low near-constant value as a point B. To define points A and B more clearly, we introduce the corresponding differential curves of the consumption curves as shown in Supplementary Fig. 19. In Supplementary Fig. 19, three solute consumption curves corresponding to Fig. 5 are plotted with their corresponding differential plots. The differential plots show the relation of consumption rate versus time. The point that the differential curve showing a minimum value and starting to go up is defined

as point A, which indicates that a remarkable change in the solution results in a turning consumption rate. The cross point where the differential curve goes down and flats assigned by using the tangent lines is defined as point B, which means the final slow down for the growth process. It should be noted that the first appearance of a visible crystal by naked eyes is around the assigned point A, showing the close relation of a growing single crystal with the acceleration in solute consumption. As for point B, the consumption of solute slows down obviously because of the depletion in supersaturation, and since the regular crystal growth comes to a stop, it marks a good timing to harvest the product before some unwanted second growth occurs.

The crystallization process of the FAPbI$_3$/(GBL + PPG-3000) solution (blue plot) is therefore divided into three stages: (1) the early growth of the crystal before point A); (2) the $\alpha$-FAPbI$_3$ single crystal rapid growth period (from point A to point B); (3) the supersaturation depletion period, featuring the crystal growth almost stops, after B. In comparison, the FAPbI$_3$/GBL solution systems without PPG (red and green curves) exhibit two different situations in the crystallization processes due to its instability in producing $\delta$-FAPbI$_3$ by-products or not. The red curve corresponds to produce only $\alpha$-FAPbI$_3$ crystals and the green curve corresponds to yield both $\alpha$-FAPbI$_3$ and $\delta$-FAPbI$_3$ impurity crystals. The crystallization processes of these two cases of the solutions without PPG are also divided into three stages as early growth, rapid growth, and solute depletion period. However, it is worthwhile to note that with adding PPG, the OIHP crystal growth are largely smoothed. By comparing the overall crystal growth time and the solute consumption vs. time function as well as its differential, it is discovered that with adding

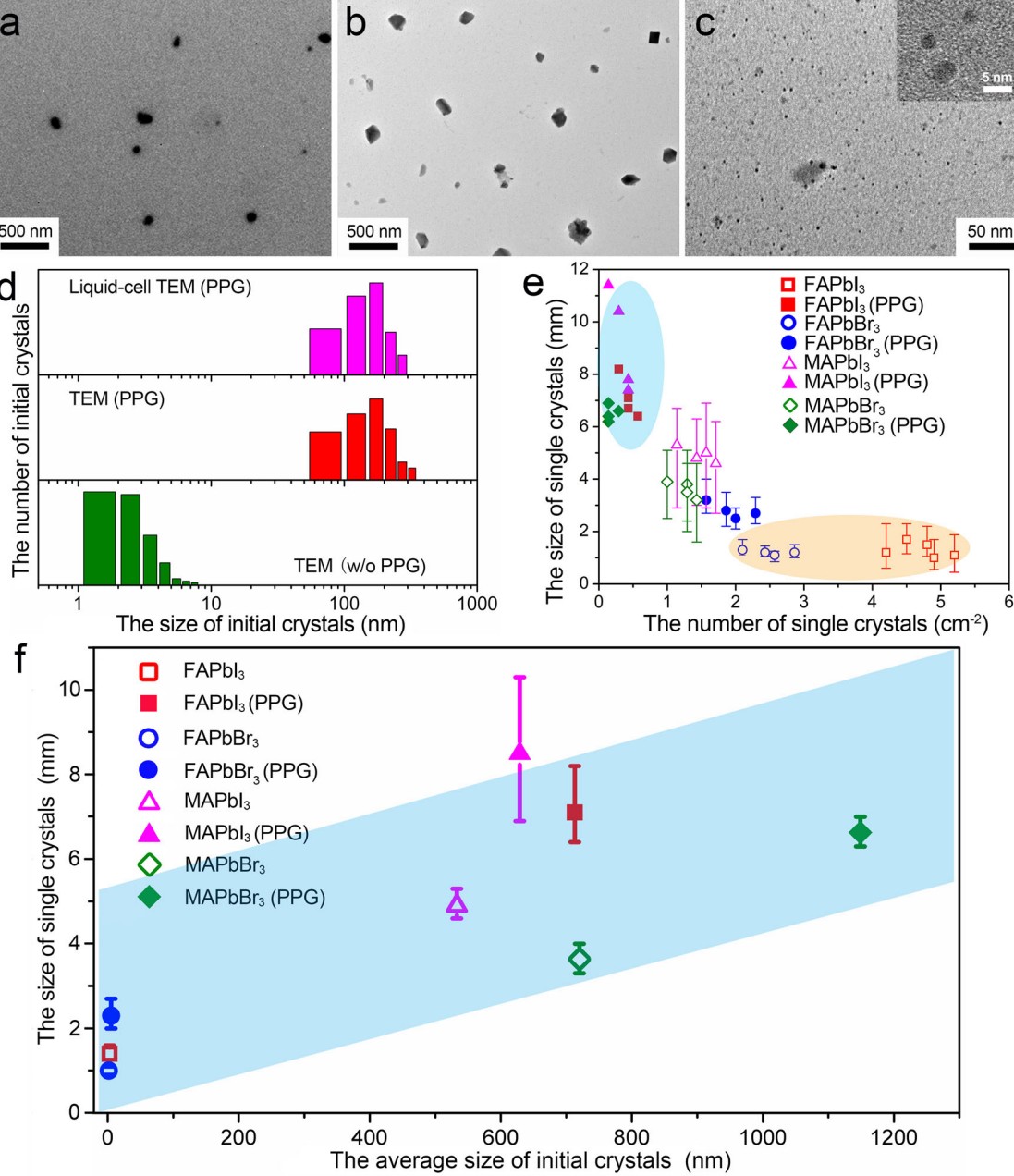

**Fig. 4 The characterization of initial nanocrystals in growth solutions at the initial growth stage. a** TEM images of FAPbI$_3$ initial nanocrystals in FAPbI$_3$/(GBL + PPG-3000) solution using a liquid cell. **b** and **c** FAPbI$_3$/(GBL + PPG-3000) solution and FAPbI$_3$/GBL solution on carbon film, respectively. **d** Size distribution of FAPbI$_3$ initial nanocrystals. **e** The correlations between the size distribution and the number of OIHPs single crystals. The error bars indicate the single crystal size range. **f** The correlations between single crystal sizes and the initial nanoparticle sizes.

PPG, the solute concentration variation vs. time for all of the early crystal growth and the rapid growth as well as the depletion of the solute are much lower than ones without PPG. The PPG largely improves the stability of the growth solution.

The solute consumed before point A mainly transforms to the observed early growth of the crystals. Supposing these crystals initially are in spherical shapes, we evaluate the crystal concentration ($N$) with the Eq. (1) as below:

$$N = \frac{n}{V} = \left(\frac{\Delta M}{\overline{m}}\right)/V = \left(\frac{\Delta C \times V}{\frac{4}{3}\pi\overline{r}^3 \times \rho}\right)/V = \frac{\Delta C}{\frac{4}{3}\pi\overline{r}^3 \times \rho} \qquad (1)$$

where $n$ is the number of initial crystals, $V$ is the growth solution volume that remains unchanged during crystallization, $\Delta M$ is the mass of solute consumption, $\overline{m}$ is the average mass of α-FAPbI$_3$

early crystals, $\overline{r}$ is the average radius of α-FAPbI$_3$ initial crystals, $\rho$ is the density of α-FAPbI$_3$ initial nanocrystals, $\Delta C$ is the solute consumption in concentration.

Combined with the TEM characterization, we can understand the effects on the nucleation process with the addition of PPG-3000 semi-quantitatively. Revealed in Fig. 4d, the average diameters of α-FAPbI$_3$ initial nanocrystals in FAPbI$_3$/(GBL + PPG) solution and FAPbI$_3$/GBL solution are 197.4 and 3.3 nm, respectively. Form Fig. 5, it can be realized that the solute consumption at point A of the blue curve is 0.11 and 0.025 g/mL of the red curve, for FAPbI$_3$/(GBL + PPG-3000) and FAPbI$_3$/GBL, respectively. The theoretical density of FAPbI$_3$ is 4.08 g/cm$^3$. Therefore, the initial nanocrystal concentrations ($N$) of FAPbI$_3$/GBL solution is calculated as ~$4.2 \times 10^{16}$/mL, which is four orders

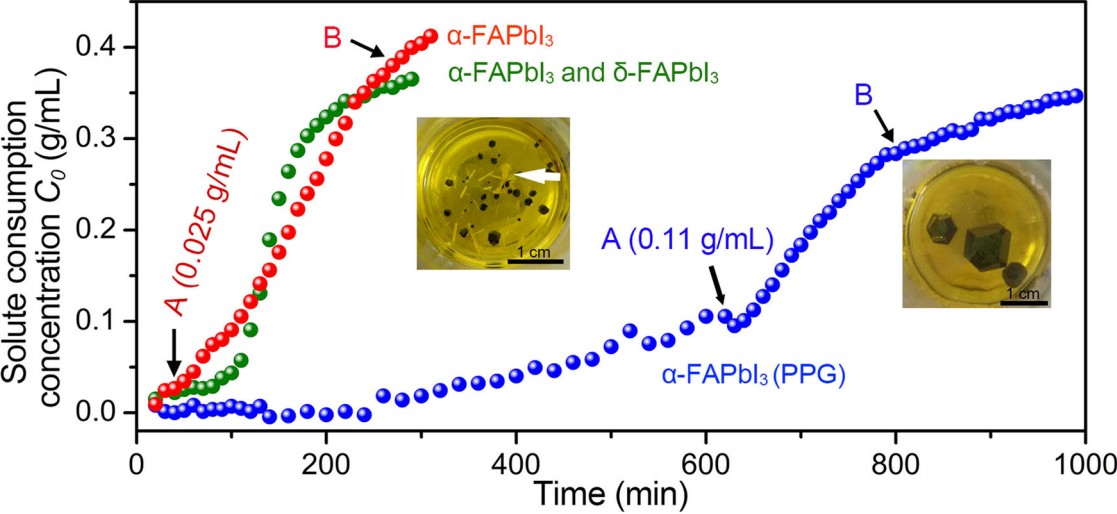

**Fig. 5 The crystallization kinetics analysis of FAPbI₃ single crystals. a** The solute consumption as a function of the growth time at 90 °C. The FAPbI₃/GBL solution ($C_{FAPbI3}$ = 0.75 g/mL): (1) only black crystals grow (red) or (2) yellow crystals also appear (green); The FAPbI₃/(GBL + PPG) solution ($C_{FAPbI3}$ = 0.75 g/ mL, $C_{PPG}$ = 0.05 g/mL) that only black crystals (blue) grow. The yellow needle-like crystals, marked by the white arrow in the photo, are δ-FAPbI₃.

of magnitude higher than that of the FAPbI₃/(GBL + PPG-3000) solution (~$8.8 \times 10^{11}$/mL). These results roughly agree with the numbers of FAPbI₃ initial nanocrystals revealed from TEM observation of ~1000/μm² to ~1.0/μm² with and without PPG-3000, respectively. Both the TEM and EIS analysis reveal that PPG-3000 significantly reduces the nucleation rate and promote the growth of large size OIHPs single crystals. This also indicates that PPG-3000 enhances the stability of growth solution at high supersaturation, thus the quality of the single crystals.

**The PC crystallization mechanisms**. We thus propose the mechanism of the PC route in Fig. 6. For the OIHPs precursor solutions, the regular solvents (GBL, DMF, or DMSO) have a coordination of oxygen which lead to form the Pb–solvent complexes at room temperature, in which, the Pb–dimethylsulfoxide (DMSO) complex has been well known[39]. For the ITC method as shown in Fig. 6a, the driving force for the OIHPs crystal's nucleation comes from the dissociation of the precursor–solvent complexes at elevated temperatures, which in turn results in a rather fast nucleation rate and great numbers of nuclei[26], which is demonstrated in Fig. 5 and our TEM observations. In contrast, as shown in Fig. 6b, the nucleation process by the PC approach is efficiently controlled by the polymer due to the coordinative interaction between the polymer and the lead polyhalide complex. The PPG occupies some of coordination of Pb (II) ions by its oxygen groups, replacing coordinating iodide ions and solvent molecules in the precursor solution. When the solution is heated to the crystal's nucleation temperature, the equilibrium of iodoplumbates complexes with PPG coordination monomers gradually shifts to dissociation and the supersaturation status emerges for nucleation of Pb–I₆ complex. The Pb–I₆ complex in turn acts as the nuclei to form nanoparticles, which eventually grow to the sizes of 100–300 nm. A few of these nanocrystals are finally activated for the further growth of the centimeter-sized single crystal. The proper polymers, such as PPG, PEG, PAA, or PVA with oxygen groups can remarkably reduce the numbers of single crystals and thus maintain the concentration and stability of solution, while other additives such as pentadecane, eicosane, polystyrene (PS), would have negative effect (Supplementary Fig. 1). This indicates that the appropriate coordinative interaction

between the oxygen groups and Pb²⁺ ions is important to ensure the solution stability for high growth rate.

In summary, we demonstrate a general route through polymer control for the growth of large-size and high-quality OIHPs single crystals. As examples, the FAPbX₃ (X = I, Br), MAPbX₃ (X = I, Br, Cl), MAPb(I/Br)₃, MAPb(Br/Cl)₃, MA/FAPbX₃ (X = I, Br), and CsPbBr₃ single crystals are synthesized. FAPbBr₃ and FA/MAPbBr₃ single crystals exhibit outstanding long carrier lifetime and carrier diffusion length. In the PC route, the applied polymers provide coordination interactions between the oxygen groups of polymers and Pb²⁺ ions. The stability of growth solution are largely improved at high supersaturation. These lead to reduced crystal nucleation rate and prevent the nucleation of impurity crystals. Large-size high-quality OIHPs single crystals can thus be grown with simultaneous high speed. These results provide new insights for the OIHPs single crystal growth.

## Methods

**Materials**. All chemical reagents were of analytical grade and used as purchased. Lead (II) acetate trihydrate (Pb(Ac)₂·3H₂O, ≥99.5%, Fuchen chemical reagent), formamidine acetate salt (FAAc, ≥99%, Aladdin), hydriodic acid (57 wt% in water, stabilized with 1.5 wt% hypophosphorous, Aladdin), hydrobromic acid (40 wt% in water, Damao chemical reagent), hydrochloric acid (36.5 wt% in water, Damao chemical reagent), methylamine (CH₃NH₂, 40 wt% in water, Fuchen chemical reagent), cesium acetate (CsAc, Tokyo Chemical Industry), γ-butyrolactone (GBL, ≥99%, Aladdin), N, N-dimethylformamide (DMF, ≥99.9%, Aladdin), dimethyl sulfoxide (DMSO, ≥99.9% Aladdin). Polypropylene glycol (PPG, Aladdin) with different molecular weights of 2000/3000 Da. Polyethylene glycol (PEG, Aladdin) with different molecular weights of 1500/2000/4000/6000 Da. Polyacrylic acid, 50 wt% aqueous solution (PAA, molecular weight 3000 Da, Aladdin) and polyvinyl alchohol (PVA, molecular weight 16000 Da, Aladdin). Polystylene (PS, molecular weight 16000 Da, Aladdin), pentadecane (Analytical Purity, Macklin), and eicosane (Analytical Purity, Macklin).

**Synthesis of perovskite powders**. FAPbX₃ (X = I, Br) powders were prepared by reacting FAAc, lead (II) acetate trihydrate (Pb(Ac)₂·3H₂O), hydriodic acid, or hydrobromic acid in the molar ratio of 1.1:1:6. A slight surplus of FAAc is applied to avoid the production of lead (II) iodide. Firstly, the Pb(Ac)₂·3H₂O was dissolved by HX (X = I, Br) solutions under stirring to obtain clear solution in a flask at 80 °C. Next, FAAc was added to the clear solution. Then the black precipitate for FAPbI₃ or the red precipitate for FAPbBr₃ was produced in the bottom after 1–2 h stirring and heating at 80 °C. Finally, the powders were collected using the Büchner funnel filtration, washed by the anhydrous ethanol for several times, and subsequently dried at 80 °C for 24 h.

MAPbX₃ (X = I, Br, Cl) powders were prepared using a similar method as that of FAPbX₃. The difference is that in the synthesis process of MAPbX₃, the

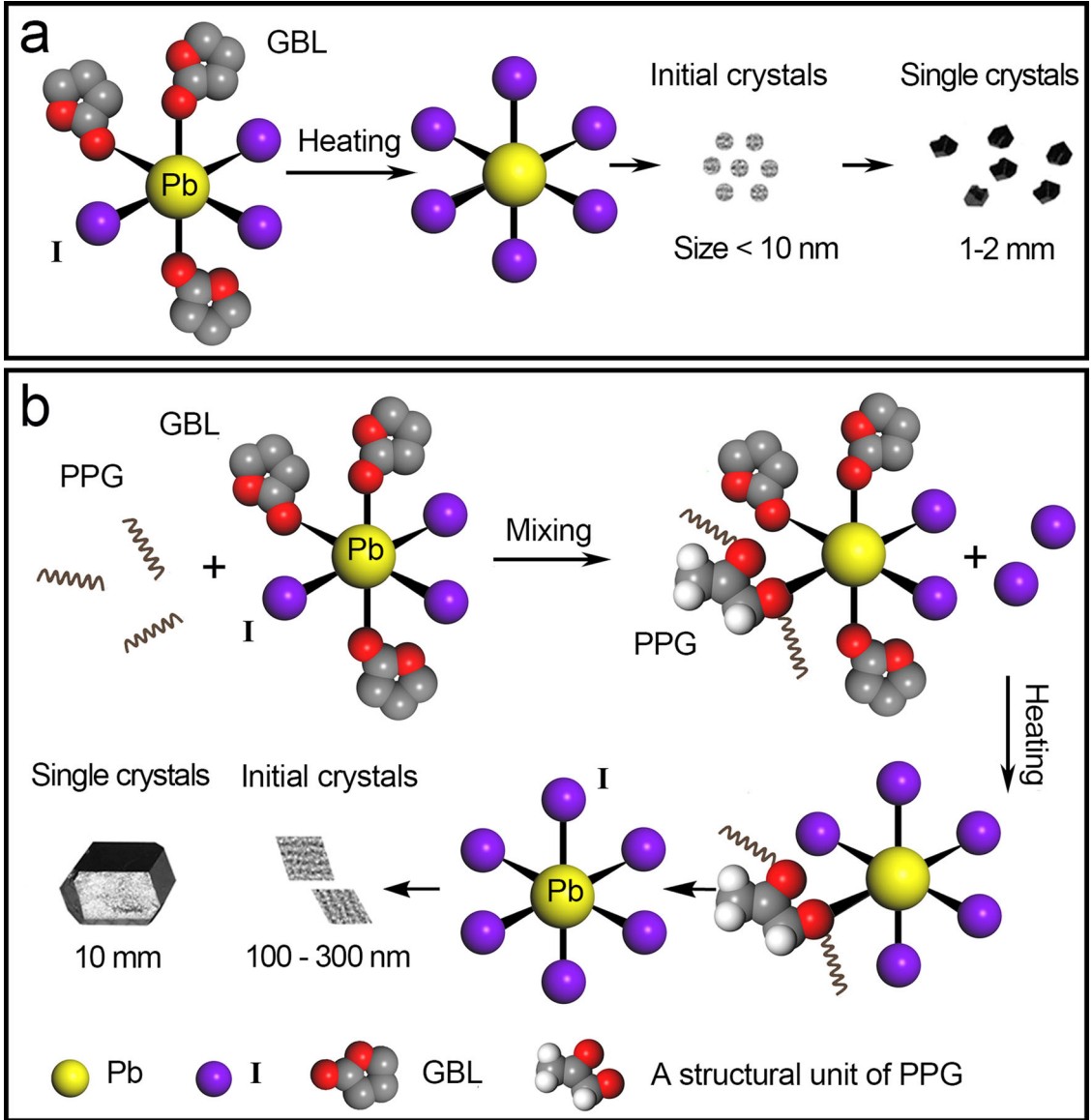

**Fig. 6 Schematic illustration of OIHPs crystallization mechanisms. a** The conventional ITC: nucleation from the dissociation of precursor–solvent complexes. **b** Our polymer-controlled nucleation process: the reaction process of the PPG with the iodoplumbates; The nucleation process is slowed down by the PPG.

methylamine (CH$_3$NH$_2$) (40 wt% in water) was used instead of the FAAc. For the MAPbCl$_3$, excessive hydrochloric acid was used to make sure lead (II) acetate trihydrate completely dissolved and the MAPbCl$_3$ powder was obtained by cooling the solution for several days. CsPbBr$_3$ powder could be prepared using cesium acetate and hydrobromic acid as an inorganic source and halogen source, respectively. The mixed-halide perovskite powders were prepared by mixing hydroiodic and hydrobromic acid or hydrobromic and hydrochloric acid as a mixed source of halogens. The mixed-organic cation perovskite powders were prepared by adding FAAc and methylamine with different ratios. Finally, the mixed MAPbI$_x$Br$_{3-x}$, MAPbBr$_x$Cl$_{3-x}$ and MA$_x$FA$_{1-x}$PbX$_3$(X = I, Br) perovskite powders were obtained.

**The crystallization of perovskite single crystals**. The obtained powders were dissolved in appropriate organic solvents with different concentrations for the growth of single crystals. FAPbI$_3$ and MAPbI$_3$ powders were dissolved in GBL for 0.75 g/mL, MAPbBr$_3$ powder was dissolved in DMF for 0.375 g/mL, FAPbBr$_3$ powder was dissolved in GBL/DMF (1:1) for 0.25 g/mL, MAPbCl$_3$ powder was dissolved in DMSO for 0.5 g/mL and CsPbBr$_3$ powder was dissolved in DMSO for 0.5 and 0.35 g/mL, respectively. After stirring for 3–4 h for a complete dissolution, a certain amount (0.01–0.1 g/mL) of polymers was added. Finally, the precursor solution was filtered using a polytetrafluoroethylene filter with a 0.2 μm pore size (Whatman) and then placed on a hot plate preheated to a certain temperature for crystallization. FAPbI$_3$, MAPbI$_3$, MAPbBr$_3$, and FAPbBr$_3$ precursor solutions were

treated at 60–95 °C for ITC crystallization. MAPbCl$_3$ and CsPbBr$_3$ precursor solutions were treated at room temperature for evaporative crystallization. The PAA-3000 (50% solution) is dried in a vacuum drying oven and subsequently ground into a powder before being added to the solution. Detail information see Supplementary Table 3.

**Characterization and measurements**. The XRD patterns of powder samples were characterized using a powder x-ray diffractometer (D8 Advance, Bruker) with Cu Kα1 radiation (λ = 1.5406 Å). UV–Vis absorption spectra of the crystals were collected on a U-3900H (Hitachi) spectrometer using an integrating sphere. The PL measurements of bulk crystals were performed with a Horiba iHR320/550 imaging spectrometer using a 325 nm Kimmon IK3301R-G laser as excitation source. Steady and transient state PL was measured using a FLS-920 fluorescence spectroscopy (Edinburgh Instruments) using 405 and 475 nm excitation wavelengths, respectively. Raman spectra were measured using a microspectroscopic Raman setup equipped with a 532 nm excitation laser (inVia Qontor, Renishaw). The FTIR spectra were analyzed by a Spectrum II infrared spectrometer in the 500–4000 cm$^{-1}$ wave number range at a resolution of 4 cm$^{-1}$. NMR spectra were collected on a Bruker 400 MHz NMR spectrometer and the spectra were analyzed in TOPSPIN. The halide compositions of the samples were carried out on a Shimadzu (XRF-1800) x-ray fluorescence (XRF) spectrometer. The electrical measurements of single crystals were carried out in air at room temperature using a Keithley 2400 digital source-meter. DLS measurements were carried out by a Zetasizer Nano ZS

(Malvern) using a 632.8 nm laser. EIS measurements of solution were performed using an electrochemistry system (Parstat 2273, Princeton Applied Research). TEM imaging and selected area electron diffraction (SAED) characterization are performed in a 200 kV FEI T20 transmission electron microscope. The liquid cell technology for TEM is performed by a Protochips holder (Poseidon 510) with amorphous silicon nitride Echips (a 550 μm × 20 μm viewing window and a 500 nm spacer). The thickness of the silicon nitride films of the observation window is 50 nm for each chip. The E-chips are processed with plasma cleaner to obtain a hydrophilic surface before loading the dilute $FAPbI_3$ growth solution.

## Data availability

The authors declare that the main data supporting the findings of this study are available within the article and its Supplementary Information files.

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

## Acknowledgements

This work was supported by the National Natural Science Foundation of China (Grant Nos. 11674015, 11674040 and 51988101) and the Fundamental Research Funds for the Central Universities (Grant No. 2018CDQYWL0048), the Beijing Outstanding Young Scientists Projects (BJJWZYJH01201910005018) and the "111" project under the grant of DB18015. We thank Prof. Fan Li at the College of Environmental and Energy Engineering, Beijing University of Technology for the discussion of the kinetics model section.

## Author contributions

Z.G.Y. and X.Y.Z. initiated and designed the project. X.D.H supervised the project. L.M. conceived the idea, developed the polymer-controlled single crystal growth method, and carried out most of experiments and measurements under the supervision of Z.G.Y. and X.Y.Z. Y.Q.P., J.H., and K.W.W. synthesized lead halide perovskite powders and carried out preliminary works. Y.P.D. took part in the OHIP synthesis for the revision of manuscript. K.W. assisted the carrier lifetime measurement. C.Q.Z. assisted the liquid cell measurement. Z.G.Y. and X.Y.Z. wrote the manuscript. All the authors reviewed and revised the manuscript.

## Competing interests
The authors declare no competing interests.
