## [Peer Review File · Nature Communications]

Reviewers' comments:

Reviewer #1 (Remarks to the Author):

In this paper, Ma et al. presented their work entitled "A Polymer Control Nucleation Route towards Unified Growth of Organic-inorganic Perovskite Single Crystals". There are quite a few issues need to be addressed before it becomes a good paper for publication.

1. Language:

a) The manuscript title does not read right. I guess the authors really meant to say "A Polymer Controlled ..."

b) Abbreviations need to follow common roles. For example, the manuscript used "polymer control (PRC)", it is probably should be shortened as (PC).

2. The manuscript states "For the first time, high quality centimeter sized MA/FAPbBr₃ single crystals were synthesized". However, there have been reports a few years ago on much larger crystals. For example, prof. Liu from China has been teaching for long time on as large as 120 mm Single-crystalline perovskite. His group even fabricated large area wafers and photodetectors, solar cells, etc. In Liu's talk, he often show dozens of single crystals for a full range of CH₃NH₃Pb(Cl_xBr_{1-x})₃ and CH₃NH₃Pb(BryI_{1-y})₃ with obviously much higher quality.

3. The manuscript claims that "It is uncovered that the PRC-nucleation process is based on an appropriate coordinative interaction between the oxygen group of polymer polypropylene glycol with Pb²⁺ ions and leads to a significant low nucleation concentration and low average nucleation rate which was dramatically decreased by up to 4 and 5 orders, respectively." First of all, the language of "dramatically decreased by up to 4 and 5 orders" is very hard to understand. It needs to be improved. Secondly, the manuscript does not seem to support "based on an appropriate coordinative interaction". The authors need to give more analyses, e.g. FTIR, NMR, etc. before they can make the claim. Pure imagination and schematic are not sufficient evidences to support the mechanism.

4. At the end of the paper, it says "The rapid and universal manufacturing of arbitrary OIHP large size and high-quality single crystals can be expected in the future." It is not scientific and should be removed.

5. In Fig. 1, the authors show photos of some single crystals. However, the dimensions and size are not clear at all. More importantly, there are plenty of defects clearly visible in the photos. How is the quality of the single crystals qualified? The authors should use high resolution X-ray rocking curve analyses to show how good their crystal quality is.

6. The manuscript has a paragraph "In summary, we demonstrate a polymer control nucleation method for the growth of whole family OIHP single crystals. High quality large size OIHP single crystals with the perfect shapes, good transparency, and excellent carrier properties are prepared with high growth rates." There are a few problems. First, I don't think the authors really meant to say "good transparency" for it is contradictory to what the perovskites are used for. Secondly, "whole family OIHP single crystals" is used without proper definition. Thirdly, in single crystal studies, "high growth rates" is not compatible with "High quality large size OIHP single crystals with perfect shapes".

Reviewer #2 (Remarks to the Author):

Perovskite single-crystal growth has drawn increasing interest and the devices from these single-crystals are currently emerging. Thus novel methods that make single-crystal growth easier are useful for simplifying the fabrication of perovskite single-crystal device. The authors showed that a material additive can increase the size of perovskite crystals grown from inverse temperature crystallization (ITC), which is the most popular crystal growth method for the perovskite community. However, the crystals grown by the demonstrated method do not exhibit significantly improved photophysical or electrical properties – such as mobility, diffusion length, TRPL decay so on.

The authors' method might be useful for obtaining big crystals, yet, there are already several publications that already show inch-sized perovskite single crystals with the ITC method. Therefore, the findings of this work are not novel enough nor represent a major advance that will

appeal to a broad audience of researchers in the community. Hence, this work may be suitable for publication in a specialized journal after a major revision.

Specific comments:

1. The authors claim in the abstract that they fabricated for the first time MA/FAPbBr₃ single-crystals in centimeter size. However, it was never a challenge for researchers to fabricate these crystals, and they were not required in any case, since large high-quality single-crystals of MAPbBr₃ and FAPbBr₃ are already easy to fabricate and are stable as well. On the other hand, for mixed MA/FAPbI₃ single crystal, which may actually have some use in device applications, there are already reports showing large crystal growth, e.g. DOI: 10.1039/C7TA04608A

3. On lines 93, 94, the authors claim the crystal growth time was reduced from a week to one day. However, this statement is misrepresentative, because the sped-up growth is not due to the addition of PPG but is a result of the authors basing their approach on the well-known ITC method. The authors are reselling the ITC method, which is already known to produce crystals in several hours.

4. The authors claim a new name for their growth method, calling it a "PRC-nucleation and crystallization approach", however, their method is essentially identical to inverse temperature crystallization (ITC). The authors merely modified ITC with an additive. The author's method is accurately described as a modified ITC method.

5. In line 102, 103 the authors claim that the crystals are of higher quality but if one checks and compares these crystals to the ones grown by ITC in Ref 22 and 24, there are no significant differences in their physical properties.

6. Figure 3a, the inset and the main figure are conveying opposite trends with respect to size with and without PPG.

7. The colloidal size and crystal sizes are very sensitive to processing conditions. They should be compared under the same temperature, concentration, purity, and age of the precursor solutions, etc. The author did not provide enough details for the reviewer to evaluate the validity of their measurements.

8. On Line 193 The authors claim the single-crystal size increases up to 5 times. I don't agree with claim, since ITC already yields crystals of the same size reported in this current manuscript. See for example DOI: 10.1016/j.mattod.2018.04.002 and 10.1002/adma.201502597

9. The authors mention that they used a mixture of solvents, yet the ratio between the two solvents is not clearly stated nor is the precise concentration of PPG given. The manuscript often states that a "certain" amount of PPG is added (see line 356 for example). The authors should refrain from using ambiguous terms and should instead give the exact concentrations.

10. Figure 5A shows that in the PPG added solution the crystallization completed within 13 h, while in the case of ITC, the completion occurred in 3 h. This observation is in clear conflict with the authors' main claim in the manuscript's introduction that the crystallization time is decreased from days to hours by the use of PPG.

11. The brand and purity of the PPG was not stated. Please state the brand, molecular weight, and purity of PPG.

12. The description of the method for growing each type of crystal is cursory. It should be expanded.

Reviewer #3 (Remarks to the Author):

Single crystals of organic-inorganic hybrid methylammonium lead trihalide perovskites show remarkably physical properties in photovoltaic applications such as high efficient photo-electric transition efficiencies, low trap density and charge transport properties. However, the growth of high-quality organic-inorganic halide perovskite semiconductor single crystals is a slow process, normally takes days or even weeks. The growth of distinctive mixtures on A or X sites for ABX₃ single crystals are also difficult, particularly, the growth of all family single crystals are impossible yet, though distinctive and various nucleation routes and growth mechanisms have been developed and proposed. The rapid growth of large size and high quality organic-inorganic

perovskite single crystals are highly valuable, particularly when recent rapid developments of these materials for applications in x-ray and γ -ray detectors and sensors emerged.

The authors reported a novel polymer control nucleation route to develop an efficient approach for large size and high quality organic-inorganic perovskite single crystal growth. They synthesized high quality centimeter sized MA/FAPbBr₃ single crystals for the first time, and further, the whole-family high-quality single crystals including lead-based perovskite, mixed organic cations (FA/MA) and halide (I/Br, Br/Cl) single crystals. The polymer control nucleation mechanisms by coordinative interaction between the oxygen group of polymer polypropylene glycol with Pb²⁺ ions are also clarified by detail kinetic researches and in situ TEM and other systematic experimental observation. In the future, large scale growth of single crystals of all family MA/FAPbX₃ can be expected.

The results are novel, very high quality and extremely important. It is suggested to publish it after minor revision:

- 1, In Figure 1 b, d, e and the related text, the x values in the MA_xFA_{1-x}PbX₃ (X = I, Br) should be denoted;
- 2, The labels of x and y axes in the insets of Fig. 2, should be reformatted. The characters are too small to be noticed;
- 3, In page 8, equation 1 has no label yet and it should be (1); the others should be (2) and (3);
- 4, Comparing to other organic-inorganic perovskite single crystals, Why the growth of large size MA/FAPbBr₃ single crystals is difficult and was firstly reported in this work;
- 5, The oxygen group of polymer polypropylene glycol plays a key role during the growth of organic-inorganic perovskite single crystals, is there other polymer with oxygen group may have the same behavior?

Reviewer #4 (Remarks to the Author):

This work reports an interesting approach to controlling polymer nucleation process for highly efficient, large size and high-quality hybrid organic-inorganic perovskite single crystals. Compared with conventional methods based on solution temperature-lowering route, anti-solvent vapor assisted crystallization and slow evaporation, the method reported here present several notable advantages such as rapid crystal growth and readily applicable to crystals of varied compositions. Overall, this work is carried out with care. I would recommend its publication only if the authors could take into account the following points in revising the manuscript:

1. In the introduction part, for the sake of a balanced overview, the authors should include the discussion of all-inorganic perovskite materials given their importance in various applications. For example, all-inorganic perovskite nanocrystals have found useful as X-ray scintillators (Nature 561, 88-93 (2018) doi:10.1038/s41586-018-0451-1).
2. With the addition of PPG, the formation of the yellow phase of δ -FAPbI₃ was inhibited. What are the mechanisms during the growth for avoiding the δ -FAPbI₃? In addition, the author should provide evidence of the δ -FAPbI₃ without the PPG.
3. In Fig. 3d, the authors believed that the variation in the intensities of the absorption is ascribed to the free iodine ions and the increased concentration of I₃⁻, derived from the iodine ions of iodoplumbates that are replaced by oxygen groups of PPG. Was the oxygen of PPG coordinated to the octahedron Pb (II) ion? I would suggest additional characterizations for the sample, such as NMR or FTIR.
4. An in-situ electrochemical impedance spectroscopy (EIS) method was applied to probe the kinetics of the solution's concentration. How was characterization performed (such as 0.02g/ml and 0.11g/ml). The details should be provided.
5. In Fig. 5a, without the addition of the PPG, alpha-FAPbI₃ and δ -FAPbI₃ are likely to form. How do the authors to distinguish one another and calculate the concentration?
6. What is the criterion for identifying point A and B? It is rather qualitative by choosing the appearance of the visible crystal.

Response to Reviewers' Comments

Reviewer #1 (Remarks to the Author):

General Comments:

In this paper, Ma et al. presented their work entitled “A Polymer Control Nucleation Route towards Unified Growth of Organic-inorganic Perovskite Single Crystals”. There are quite a few issues need to be addressed before it becomes a good paper for publication.

Reply:

Thanks for the reviewer’s constructive comments. We address the reviewer’s comments by supplementing more systematic experiments and analyses. Correspondingly, we re-write many parts of the main text, including revising the manuscript’s title and abstract. With these, we believe that the current version meets the high standard of Nature Communications.

Comment 1:

Language:

- a) The manuscript title does not read right. I guess the authors really meant to say “A Polymer Controlled ...”
- b) Abbreviations need to follow common roles. For example, the manuscript used “polymer control (PRC)”, it is probably should be shortened as (PC).

Reply:

The title of the manuscript is revised as “**A Polymer Controlled Nucleation Route towards the Generalized Growth of Organic-Inorganic Perovskite Single Crystals**”. The same change is also applied in the main text.

Following the reviewer’ suggestions, the abbreviation of “polymer controlled” is changed as “**PC**”.

Comment 2:

The manuscript states “For the first time, high quality centimeter sized MA/FAPbBr₃ single crystals were synthesized”. However, there have been reports a few years ago on much larger crystals. For example, prof. Liu from China has been teaching for long time on as large as 120 mm Single-crystalline perovskite. His group even fabricated large area wafers and photodetectors, solar cells, etc. In Liu’s talk, he often show dozens of single crystals for a full range of CH₃NH₃Pb(Cl_xBr_{1-x})₃ and CH₃NH₃Pb(Br_yI_{1-y})₃ with obviously much higher quality.

Reply:

Figure R1. (Supplementary Figure S11) (a, c) The XRD patterns of $\text{MA}_{0.16}\text{FA}_{0.84}\text{PbBr}_3$ and $\text{MA}_{0.33}\text{FA}_{0.67}\text{PbBr}_3$ single crystals and the corresponding X-ray rocking curves for the (001) planes. (b, d) PL time-decay trace of $\text{MA}_{0.16}\text{FA}_{0.84}\text{PbBr}_3$ and $\text{MA}_{0.33}\text{FA}_{0.67}\text{PbBr}_3$ crystals after 405 nm excitation.

Thanks for the referee's comments. We fully agree and appreciate that Prof. Liu and coworkers have done outstanding researches in synthesizing organic-inorganic halide perovskites (OIHPs) single crystals. In this work, we report a polymer-controlled nucleation crystal growth approach towards a general route for synthesizing OIHPs single crystals. A series of large-size high-quality MA/FAPbBr₃ single crystals are grown which have not been reported in literatures as far as we know. These single crystals also have interesting physical properties. As illustrated in **Fig. R1**, (**Supplementary Fig. S11**) the carrier lifetime of $\text{MA}_{0.16}\text{FA}_{0.84}\text{PbBr}_3$ and $\text{MA}_{0.33}\text{FA}_{0.67}\text{PbBr}_3$ single crystals reaches 8712 ns and 8420 ns, respectively. These progresses are highly desirable in the optoelectronic application of OIHP single crystals.

According to the suggestion of the reviewer, we revise this sentence as “**High-quality centimeter-sized MA/FAPbBr₃ single crystals are synthesized with outstanding carrier lifetime.**” in the Abstract. At the meantime, we appreciate and discuss Prof. Liu's contribution in the revised manuscript as below:

The growth of large-size high-quality perovskite single crystals has drawn increasing interest. Many significant progresses have been made in the growth of OIHPs single crystals. For example, MAPbI₃ single crystal with size up to 71 mm with aid of a seed-induced heterogeneous nucleation was fabricated.^[20] and a low-temperature-gradient crystallization method for high-quality MAPbBr₃ single crystals was developed.^[21] Line 49-53, Page 1.

Comment 3:

The manuscript claims that “It is uncovered that the PRC-nucleation process is based on an appropriate coordinative interaction between the oxygen group of polymer polypropylene glycol with Pb^{2+} ions and leads to a significant low nucleation concentration and low average nucleation rate which was dramatically decreased by up to 4 and 5 orders, respectively.” First of all, the language of “dramatically decreased by up to 4 and 5 orders” is very hard to understand. It needs to be improved. Secondly, the manuscript does not seem to support “based on an appropriate coordinative interaction”. The authors need to give more analyses, e.g. FTIR, NMR, etc. before they can make the claim. Pure imagination and schematic are not sufficiently evidence to support the mechanism.

1. About the language of “dramatically decreased by up to 4 and 5 orders”:

Reply:

Thanks for the reviewer’s comments and suggestions.

To avoid confusion and clarify the meaning the sentence, “...and leads to a significant low nucleation concentration and low average nucleation rate which was dramatically decreased by up to 4 and 5 orders, respectively.” is revised as “**Through the polymer control nucleation process, the average nuclei concentration can be decreased by 4 orders of magnitude.**” in the abstract. Line 32-33, Page 1.

2. About the “appropriate coordinative interaction”, the discussion on coordination interaction is also required by **Reviewer 4 Comment 3.**

Here, we reply these two comments together.

Reply: Regarding the coordination interaction, as suggested by the reviewer, we conduct a series of experiments to prove the coordinative interaction between Pb^{2+} and PPG-3000 by Raman and Fourier Transformed Infrared Spectroscopy (FTIR). The details of the experiments are described and explained below.

The samples used for characterization:

For the Raman measurements, we directly mix α -FAPbI₃ powders with PPG-3000 (liquid) to prove the formation of Pb-O coordination bonds. For the FTIR spectra investigation, the PPG-3000 and PEG-200 (liquid) are selected as solvent and the α -FAPbI₃/PPG-3000 powders and α -FAPbI₃/PEG-200 solutions are prepared.

The details of results are shown below:

(1) Raman Spectra

Figure R2. (Supplementary Figure S13) Raman spectra of PPG-3000 (black), α -FAPbI₃ powders (red), Pb(Ac)₂ powders (green), α -FAPbI₃/PPG-3000 powders (blue) using a 532 nm wavelength laser.

The Raman spectra of PPG-3000, α -FAPbI₃ powders, PbAc₂ powders, and FAPbI₃/PPG-3000 powders are shown in **Fig. R2**. There is no peak in the Raman spectrum of pure PPG-3000 at the range of 50-600 cm⁻¹. The peak at 132 cm⁻¹ of α -FAPbI₃ powders corresponds to the Pb-I bonding [R1]. The Pb-O bonding of Pb(Ac)₂ powders is observed at 216 cm⁻¹ [R2]. For the FAPbI₃/PPG-3000 powders, a new peak appears at 210 cm⁻¹ compared with the Raman spectrum of the α -FAPbI₃ powder. The new peak is assigned to Pb-O bond [R2] which corresponds to and agree with the Raman spectrum of PbAc₂ powders.

(2) FT-IR spectra

Figure R3. (Supplementary Figure S14) (a) The IR spectra of α -FAPbI₃ dissolved in PEG-200. (b) The band near 1720 cm⁻¹ in the IR spectra. (c, d) The band near 1100 cm⁻¹ in the IR spectra as measured, and with the deconvoluted components. (e) The IR spectra of PPG-3000 and FAPbI₃/PEG-3000 solution. (f) The band near 1100 cm⁻¹ in the IR spectra with different concentration of FAPbI₃.

The α -FAPbI₃/PEG-200 solution and α -FAPbI₃/PPG-3000 solution with different α -FAPbI₃ concentration are measured by FTIR, respectively. (Fig. R3) The C-O-C band (~1100 cm⁻¹) and C-O-H band (~1060 cm⁻¹) [R3] shift towards lower frequency, due to the formation Pb-O coordination bonds.

As shown in Fig. R3c, d and Table R1 (Supplementary Table 6), after dissolving FAPbI₃ powder into PEG-200 (1.13 g/mL), the 1102.1 cm⁻¹ peak shifts to a lower frequency of 1093.3 cm⁻¹, which can be assigned to the $\nu(\text{C-O-C})_{\text{trans}}$. [R4] Furthermore, the 1066.7 cm⁻¹ peak shifts to a lower frequency of 1060.4 cm⁻¹, which is assigned to the $\nu(\text{C-O-H})$. As reported in literature, [R4] the shift in the peak towards lower frequency for PEG is attributed to the binding of C-O-C and C-O-H groups with metal ions, which reduces the electron density on O and weakens the C-O bond in turn, leading to a lowered vibration frequency. [R4]

Table R1. (Supplementary Table 6) Parameters of the single components obtained after deconvolution of the band near 1100 cm^{-1} in liquid PEG-200 and liquid PEG-200/FAPbI₃ (1.13 g/mL).

	PEG-200 (cm^{-1})	α -FAPbI ₃ /PEG-200 (cm^{-1})
$\nu(\text{C-C})$	1034.4	1032.9
$\nu(\text{C-O-H})$	1066.7	1060.4
$\nu(\text{C-O-C})$ trans	1102.1	1093.3
$\nu(\text{C-O-C})$ gauche	1124.7	1125.2
$\delta(-\text{CH}_2-)$	1143.2	1143.0

To confirm these, we further use the PPG-3000 as solvent to dissolve the FAPbI₃ powder. It is found that the solubility of FAPbI₃ in PPG-3000 is low, which does not exceed 0.1 g/mL. We first add excess FAPbI₃ powders into the PPG-3000 and stir for 12 hours, and then filter it to get a saturated FAPbI₃/PPG-3000 solution. The FAPbI₃/PPG-3000 saturated solution is diluted by PPG-3000 to obtain different concentration of FAPbI₃/PPG-3000 solution for FTIR measurements. As shown in Fig. R3e – 3f, the experiment results are similar with the above-mentioned results of PEG-200.

The coordination of glycol polymers with Pb²⁺ are thus proved by these combined experiments of Raman and IR spectra. The corresponding experimental results are supplemented in SI.

Comment 4:

At the end of the paper, it says “The rapid and universal manufacturing of arbitrary OIHP large size and high-quality single crystals can be expected in the future.” It is not scientific and should be removed.

Reply:

The sentence is removed.

Comment 5:

In Fig. 1, the authors show photos of some single crystals. However, the dimensions and size are not clear at all. More importantly, there are plenty of defects clearly visible in the photos. How is the quality of the single crystals qualified? The authors should use high resolution X-ray rocking curve analyses to show how good their crystal quality is.

Reply:

Thanks for the reviewer’s comments.

(1) For the size of the single crystals

A block in the coordinate paper in Fig. 1 is for 1×1 mm. The size details of these single crystals are shown in Table R2 (Supplementary Table 1). Some of the dark corners

or lines are the projected images of the crystals or the transmitted line images of the coordinate papers under the crystal.

Table R2 (**Supplementary Table 1**) The sizes of the OIHPs single crystals in Figure 1 (Main text).

OIHPs single crystal	FAPbI ₃	FAPbBr ₃	CsPbBr ₃
Size: L×W (mm×mm)	11×10	8×8	10×3
OIHPs single crystal	MAPbBr ₃	MAPbCl ₃	MAPbI ₃
Size: L×W (mm×mm)	8×6	5×5	12×10
OIHPs single crystal	MAPbI _{2.1} Br _{0.9}	MAPbI _{1.6} Br _{1.4}	MAPbI _{0.12} Br _{2.88}
Size: L×W (mm×mm)	10×8	5×5	8×8
OIHPs single crystal	MAPbBr _{2.85} Cl _{0.15}	MAPbBr _{2.46} Cl _{0.54}	MAPbBr _{2.1} Cl _{0.9}
Size: L×W (mm×mm)	8×6	9×8	9×7
OIHPs single crystal	MA _{0.57} FA _{0.43} PbI ₃	MA _{0.31} FA _{0.69} PbI ₃	MA _{0.26} FA _{0.74} PbI ₃
Size: L×W (mm×mm)	7×5	9×7	10×9
OIHPs single crystal	MA _{0.11} FA _{0.89} PbI ₃	MA _{0.86} FA _{0.14} PbBr ₃	MA _{0.76} FA _{0.24} PbBr ₃
Size: L×W (mm×mm)	4×4	10×10	7×7
OIHPs single crystal	MA _{0.63} FA _{0.37} PbBr ₃	MA _{0.45} FA _{0.55} PbBr ₃	
Size: L×W (mm×mm)	4×4	4×4	

(2) The X-Ray Rocking Curves and time resolved photoluminescence (TRPL) referring to **Fig. R4, R5, R6, R7** are supplemented.

Following the reviewer's comments and suggestions, we conduct X-ray rocking curve investigations. The X-ray rocking curves of several single crystals are shown in **Fig. R4-R6**. The lowest FWHM values of the FAPbI₃, MAPbI₃, FAPbBr₃ and MAPbBr₃ single crystalline samples are measured as 0.118°, 0.236°, 0.174° and 0.127°, respectively which indicate that these single crystals are of high crystallinity without impurity crystals, defect boundary or high density of dislocation.

Furthermore, the long carrier lifetime of FAPbBr₃ reaches 10199 ns (**Fig. R7**) which further demonstrates the single crystals' high quality of the crystallinity with low defect density. The carrier lifetime of other six FAPbBr₃ single crystals are also over 6000 ns. (**Supplementary Table S4**)

Figure R4. (Supplementary Figure S5) (a) The FAPbI₃ single crystal that the crystallization process is assisted by PAA-3000. (b-d) The XRD patterns of FAPbI₃ single crystal on 1, 2, 3 facets and the corresponding X-ray rocking curves for the (001), (001) and (011) planes, respectively.

Figure R5. (Supplementary Figure S6) (a-d) The XRD patterns of FAPbI₃ and MAPbI₃ single crystals on different facets and the corresponding X-ray rocking curves for the (011) planes of FAPbI₃ and the (112) planes of MAPbI₃. The crystallization process is assisted by PPG-3000.

Figure R6. (Supplementary Figure S7) (a-d) The XRD patterns of MAPbBr₃ and FAPbBr₃ single crystals and the corresponding X-ray rocking curves for the (001) planes of MAPbBr₃ and FAPbBr₃.

Figure R7. (a) PL time-decay trace of FAPbBr₃ crystals monitored at $\lambda = 590$ nm with 405 nm excitation, with biexponential fit revealing fast ($\tau_1 = 2493$ ns) and slow ($\tau_2 = 10199$ ns) components; (b) PL time-decay traces of the other six FAPbBr₃ crystals monitored at $\lambda = 590$ nm with 405 nm excitation.

Comment 6:

The manuscript has a paragraph “In summary, we demonstrate a polymer control nucleation method for the growth of whole family OIHP single crystals. High quality large size OIHP single crystals with perfect shapes, good transparency, and excellent carrier properties are prepared with high growth rates.” There are a few problems. First, I don’t think the authors really meant to say “good transparency” for it is contradictory to what the perovskites are used for. Secondly, “whole family OIHP single crystals” is used without proper definition. Thirdly, in single crystal studies, “high growth rates” is not compatible with “High quality large size OIHP single crystals with perfect shapes”.

Reply:

Thanks for the reviewer's comments.

(1) About the “good transparency”

In many cases, OIHP single crystals are for non-transparence applications such as x-ray detection etc. While, transparency is a new physical property of OIHP crystal which have potential optical applications and will open new applications and research fields. We revise the text as: **It can be noticed that the Br based single crystals show good transparency. These transparent OIHPs crystals may provide potential applications in optical devices.** Line 90-92, Page 3.

(2) About “whole family OIHP single crystals”

Thanks for the reviewer's comments. We accept the reviewer's comment and agree that there will be still many OIHP members to be investigated and developed. We revise the “whole family OIHP single crystals” as **“The synthesis of three series OIHPs of simple, mixed-halide and mixed-organic cation single crystals demonstrates the capability and flexibility of this PC route.”** Line 92-94, Page 3.

(3) About the relation of “high growth rate” and the “high quality”

Thanks for the reviewer's comments. We fully agree with the referee that in the crystallization process, large-size high-quality single crystals are usually hard to be obtained under high growth rate. Rapid growth usually destroys the stability of growth solution which normally could lead to the formation of new crystals and defect structures.

Indeed, this is exactly one of the novities of the current research for solving the dilemma of **“high growth rate” and the “high quality”** by PC route. Through PC route, the stability of organic growth solutions is effectively improved by adding polymers. With the largely improved stability of the growth solution, the spontaneous nucleation and defect structure are avoided. Thus, high-quality single crystals are grown with high growth rate. This shed lights for a new strategy of growing OIHP single crystals with high quality yet high speed. Please refer to the discussion on the growth rate and carrier lifetime of the OIHPs single crystals through Line 122-127, Page 3 and Fig.2d.

Reviewer #2 (Remarks to the Author):

General Comments:

Perovskite single-crystal growth has drawn increasing interest and the devices from these single-crystals are currently emerging. Thus novel methods that make single-crystal growth easier are useful for simplifying the fabrication of perovskite single-crystal device. The authors showed that a material additive can (1) increase the size of perovskite crystals grown from inverse temperature crystallization (ITC), which is the most popular crystal growth method for the perovskite community. (2) However, the crystals grown by the demonstrated method do not exhibit significantly improved photophysical or electrical properties such as mobility, diffusion length, TRPL decay so on. The authors' method might be useful for obtaining big crystals, yet, there are already several publications that already show inch-sized perovskite single crystals with the ITC method. Therefore, the findings of this work are not novel enough nor represent a major advance that will appeal to a broad audience of researchers in the community. Hence, this work may be suitable for publication in a specialized journal after a major revision.

Reply:

Thanks for referee's constructive comments of pointing out **the importance of growing large-size and high-quality perovskite single crystal**.

The main concerns from referee are the physical properties of the grown single crystals. Though it is not the aim of the current research, in the revised version, we supplement some examples to demonstrate the outstanding and unique physical properties such as TRPL decay time of single crystals grown via the PC route.

The examples are show in the revised **Fig. 2** and **Table 1**, Supplementary **Fig. S10, S11** and **Table S4**. It demonstrates the outstanding physical properties of TRPL decay time of synthesized single crystals via the PC route. As reported in literatures, TRPL decay time is critical for the performance of ultraviolet-to-infrared photodetectors and x-ray, γ -ray detectors. ^[R5-R8] As shown in **Fig. R7 (Fig. 2a** in main text), the carrier lifetime of FAPbBr₃ by PC route can reach 10199 ns, which is **over four times** than that of FAPbBr₃ by regular synthesizing routes^[R9] and **one order longer** than other ABX₃ (A=Cs, MA, FA; X=I, Br, Cl) single crystals as reported in literatures. ^[R5, R9-R16]

Figure R7. (a) PL time-decay trace of FAPbBr₃ crystals monitored at $\lambda = 590$ nm with 405 nm excitation, with biexponential fit revealing fast ($\tau_1 = 2493$ ns, violet curve) and slow ($\tau_2 = 10199$ ns, blue curve) components; (b) PL time-decay traces of the other six FAPbBr₃ crystals monitored at $\lambda = 590$ nm with 405 nm excitation. (Illustrated again in this file.)

Specific comments:

Comment 1:

The authors claim in the abstract that they fabricated for the first time MA/FAPbBr₃ single-crystals in centimeter size. However, it was never a challenge for researchers to fabricate these crystals, and they were not required in any case, since large high-quality single-crystals of MAPbBr₃ and FAPbBr₃ are already easy to fabricate and are stable as well. On the other hand, for mixed MA/FAPbI₃ single crystal, which may actually have some use in device applications, there are already reports showing large crystal growth, e.g. DOI: 10.1039/C7TA04608A

Reply:

Thanks for the reviewer's comments and bring us the attention of the reference that MA/FAPbI₃ single crystal has been applied in some devices (DOI: 10.1039/C7TA04608A).

We report the fabrication of MA/FAPbBr₃ single-crystals in centimeter size which has not been reported in the community yet. As mentioned by the reviewer, “novel methods that make single-crystal growth easier are useful for simplifying the fabrication of perovskite single-crystal device.” Indeed, high quality and novel physical properties are highly required in the community in addition to the large size OIHP single crystals. As the example mentioned by the reviewer, “**large high-quality single-crystals of MAPbBr₃ and FAPbBr₃ are already easy to fabricate and are stable as well**”. However, the longest carrier lifetime of MAPbBr₃ and FAPbBr₃ single crystal was 2272 ns [R9] and much longer carrier lifetime has been highly required. Herein, the FAPbBr₃ single crystal prepared through PC route shows a breakthrough along this trend. **The carrier lifetime of FAPbBr₃ by PC method reaches 10199 ns**, which is much higher than the reported values (See Fig. 2 in Main text). A long carrier lifetime is one of the critical physical properties for fabricating highly efficient X-ray and photoelectron detectors.

As shown in **Fig. R1 (Supplementary Fig. S11)**, the carrier lifetimes of $\text{MA}_{0.16}\text{FA}_{0.84}\text{PbBr}_3$ and $\text{MA}_{0.33}\text{FA}_{0.67}\text{PbBr}_3$ single crystals reach 8712 ns and 8420 ns, respectively. In comparison, the carrier lifetime of the mixed MA/FAPbI₃ single crystals is about 1000 ns. ^[R10] For instance, the $\text{MA}_{0.9}\text{FA}_{0.1}\text{PbI}_3$ single crystal has the longest carrier lifetime in this series, which is 1075 ns (**Table R3**). Therefore, the MA/FAPbBr₃ single crystals fabricated in the current study exhibit interesting properties and the growth of MA/FAPbBr₃ single crystal is important for the fabrication of high-performance optoelectronic devices, proving the significance of this work.

Figure R1. (Supplementary Figure S11) (a, c) The XRD patterns of $\text{MA}_{0.16}\text{FA}_{0.84}\text{PbBr}_3$ and $\text{MA}_{0.33}\text{FA}_{0.67}\text{PbBr}_3$ single crystals and the corresponding X-ray rocking curves for the (001) planes. (b, d) PL time-decay trace of $\text{MA}_{0.16}\text{FA}_{0.84}\text{PbBr}_3$ and $\text{MA}_{0.33}\text{FA}_{0.67}\text{PbBr}_3$ crystals after 405 nm excitation.

Table R3. The carrier lifetime of MA/FAPbBr₃ single crystals we prepared and MA/FAPbI₃ single crystals reported in literatures. ^[R10]

OIHPs	Carrier lifetime (ns)	
	slow component	fast component
$\text{MA}_{0.16}\text{FA}_{0.84}\text{PbBr}_3$	8712	2356
$\text{MA}_{0.33}\text{FA}_{0.67}\text{PbBr}_3$	8420	2750
$\text{MA}_{0.9}\text{FA}_{0.1}\text{PbI}_3$	1075	122
$\text{MA}_{0.85}\text{FA}_{0.15}\text{PbI}_3$	926	88

Comment 2: (missing from the letter)

Comment 3:

On lines 93, 94, the authors claim the crystal growth time was reduced from a week to one day. However, this statement is misrepresentative, because the speed-up growth is

not due to the addition of PPG but is a result of the authors basing their approach on the well-known ITC method. The authors are reselling the ITC method, which is already known to produce crystals in several hours.

Reply:

Thanks for the reviewer's comments. The comments of 3 and 10 by the reviewer are about the speed-up growth of the OIHPs single crystals. We accept these comments. We would like to clarify it herein: This PC route is not to further increase the growth speed of ITC, but to produce high quality OIHP single crystals in a general approach and simultaneously maintain the high growth speed. At the meantime of high growth rate, PC route can effectively decrease the nucleation rate and reduce the defects and/or impurity phase resulted from the high growth speed. With the well-controlled nucleation process through PC, high quality OIHP single crystals can be achieved as demonstrated in **Fig. 1-2** in the main text by FAPbBr₃, FAPbI₃ and the mixed MA/FAPbBr₃ single crystals.

Comment 4:

The authors claim a new name for their growth method, calling it a "PRC-nucleation and crystallization approach", however, their method is essentially identical to inverse temperature crystallization (ITC). The authors merely modified ITC with an additive. The author's method is accurately described as a modified ITC method.

Reply:

Thanks for the referee's comments. The PC route is to add polymer(s) to a solution nucleation system for single crystal growth. It is applicable to ITC method which is efficient to grow OIHP crystals but also applicable to other crystal nucleation routes from solutions (non-ITC method) for OIHP single crystal growth in addition to ITC method. The term "PC-nucleation and crystallization approach" directly reflects the key mechanism of the OIHP crystal growth. We wish you agree that the "PC" term better reflects its intrinsic characters.

Comment 5:

In line 102, 103 the authors claim that the crystals are of higher quality but if one checks and compares these crystals to the ones grown by ITC in Ref 22 and 24, there are no significant differences in their physical properties.

Reply:

Thanks for the reviewer's comments and raise our attention to the two references of Ref. 22 and 24 (Reference 26 and 27 in the revised manuscript). **Please see the revised Table 1 and Fig. 2 below as well as the revised main text (Line 110-127, Page 3)**, for a comparison of the physical properties of our work with the two references. As reported in these two references, the carrier lifetime (slow component) of MAPbBr₃ and FAPbBr₃ can reach ~300 ns (**Reference 26** in the revised manuscript) and ~2272 ns (**Reference 27** in the revised manuscript), respectively. In our work, the carrier

lifetime (slow component) of MAPbBr₃ and FAPbBr₃ reach ~923 ns and ~10199 ns which is over 3 times and 4 times of the values in references, respectively as shown in **Fig. R7 below and Supplementary Figure S10**. Furthermore, the carrier lifetime of other six FAPbBr₃ single crystals are also over 6000 ns (**Supplementary Table S4**). These crystals with longer carrier lifetime (τ) are beneficial for enhancing the performance of photodetectors.

Figure R7. (a) PL time-decay trace of FAPbBr₃ crystals monitored at $\lambda = 590$ nm with 405 nm excitation, with biexponential fit revealing fast ($\tau_1 = 2493$ ns) and slow ($\tau_2 = 10199$ ns) components; (b) PL time-decay traces of the other six FAPbBr₃ crystals monitored at $\lambda = 590$ nm with 405 nm excitation. (Illustrated again in this file.)

Figure 2 | Carrier lifetime measurements. Steady and transient state photoluminescence (PL) is measured using a 405 and 475 nm excitation wavelength, respectively. PL decay curves of a, the FAPbBr₃ single crystal and b, the FAPbI₃ single crystal. c, Comparison of carrier lifetimes of OIHPs single crystals (seven FAPbBr₃ samples were tested). (ref.

27, 28, 29 in the main text) d, The correlations between carrier lifetimes and average single crystal growth rates (in volume) based on our work and literatures. (ref. 21, 26, 27, 28 in the main text)

Comment 6:

Figure 3a, the inset and the main figure are conveying opposite trends with respect to size with and without PPG.

Reply:

Thanks for the reviewer. We apologize for this mistake. It is corrected in our revised version. (Fig.3a in revised manuscript)

Comment 7:

The colloidal size and crystal sizes are very sensitive to processing conditions. They should be compared under the same temperature, concentration, purity, and age of the precursor solutions, etc. The author did not provide enough details for the reviewer to evaluate the validity of their measurements.

Reply:

In our study, the growth processes of each type of OIHPs single crystals are repeated several times to demonstrate this generalized growth method. The typical crystal and colloidal sizes with the same synthesis conditions are compared. We also measured the colloidal sizes in the typical OIHPs growth solutions at different crystallization stages. The temperature and precursor concentration are shown in Table R4.

Table R4. The typical synthesis conditions for the OIHPs single crystals

OIHPs	Temperature	Concentration of precursor	Concentration of PPG-3000
FAPbI ₃	90 °C	0.75 g/mL	N/A
FAPbI ₃	90 °C	0.75 g/mL	0.05 g/mL
FAPbBr ₃	60 °C	0.3 g/mL	N/A
FAPbBr ₃	60 °C	0.3 g/mL	0.1 g/mL
MAPbI ₃	90 °C	0.75 g/mL	N/A
MAPbI ₃	90 °C	0.75 g/mL	0.03 g/mL
MAPbBr ₃	90 °C	0.375 g/mL	N/A
MAPbBr ₃	90 °C	0.375 g/mL	0.03 g/mL

We sample the supernatant of FAPbI₃/GBL and FAPbI₃/(GBL+PPG-3000) growth solutions at different growth stages and measure the samples by the laser dynamic light scattering. The typical results are shown in Fig. R8 and Fig. R9. The results indicate that the particle size in the FAPbI₃/GBL and FAPbI₃/(GBL+PPG-3000) growth solutions at different growth stages are similar. The FAPbI₃ nanoparticles with several nanometer and hundreds nanometer size are observed in FAPbI₃/GBL and FAPbI₃/(GBL+PPG-3000) solutions, respectively. Meanwhile, the sizes obtained by

TEM using a liquid cell technology agree with those obtained by laser dynamic light scattering.

Figure R8. The DLS size distributions by intensity of particles in the FAPbI₃/GBL growth solution every 1 hour after placing on the preheated hot plate for 20 mins.

Figure R9. The DLS size distribution by intensity of particles in the FAPbI₃/(GBL+PPG-3000) growth solution every 3 h after placing on the preheated hot plate for 20 mins.

Comment 8:

On Line 193 The authors claim the single-crystal size increases up to 5 times. I don't agree with claim, since ITC already yields crystals of the same size reported in this

current manuscript. See for example DOI: 10.1016/j.mattod.2018.04.002 and 10.1002/adma.201502597

Reply:

Thanks for the reviewer for pointing out this issue. We notice and appreciate the related papers carefully and discuss these papers in the revised manuscript. In our study, we make the comparison of the single crystal synthesized with and without the same glycol polymer while keeping the remaining condition same. This comparison is important to reveal the effect of the polymers. “5 times” here is not related to any data in the literatures.

Comment 9:

The authors mention that they used a mixture of solvents, yet the ratio between the two solvents is not clearly stated nor is the precise concentration of PPG given. The manuscript often states that a “certain” amount of PPG is added (see line 356 for example). The authors should refrain from using ambiguous terms and should instead give the exact concentrations.

Reply:

Thank for pointing out this issue. We add the details in **Table R5 (Supplementary Table S3)**.

Table R5. (Supplementary Table S3). A more detailed description of the growth conditions for each type of OIHPs single crystals.

OIHPs	C _{OIHPs} (g/mL)	Solvent	Polymer	C _{Polymer} (g/mL)	Temperature (°C)
FAPbI ₃	0.75	GBL	PPG-3000	0.05	90
FAPbBr ₃	0.5	GBL/DMF (v:v = 1:1)	PEG-1500	0.05	60
MAPbI ₃	0.75	GBL	PPG-3000	0.03	95
MAPbBr ₃	0.6	DMF	PPG-3000	0.01	62
MAPbCl ₃	0.5	DMSO	PPG-3000	0.1	25
CsPbBr ₃	0.35	DMSO	PPG-3000	0.08	25
MAPbI _{2.1} Br _{0.9}	0.75	GBL/DMF (v:v = 1:1)	PPG-3000	0.02	90
MAPbI _{1.6} Br _{1.4}	0.75	GBL/DMF (v:v = 1:1)	PPG-3000	0.1	90
MAPbI _{0.12} Br _{2.88}	0.3	GBL/DMF (v:v = 1:1)	PPG-3000	0.1	90
MAPbBr _{2.85} Cl _{0.15}	0.55	DMF	PPG-3000	0.01	60
MAPbBr _{2.46} Cl _{0.54}	0.40	DMF	PPG-3000	0.01	60
MAPbBr _{2.1} Cl _{0.9}	0.37	DMF	PPG-3000	0.01	60
MA _{0.57} FA _{0.43} PbI ₃	0.75	GBL	PPG-3000	0.06	90
MA _{0.31} FA _{0.69} PbI ₃	0.75	GBL	PPG-3000	0.06	90
MA _{0.26} FA _{0.74} PbI ₃	0.75	GBL	PPG-3000	0.1	85
MA _{0.11} FA _{0.89} PbI ₃	0.75	GBL	PPG-3000	0.1	85

Comment 10:

Figure 5A shows that in the PPG added solution the crystallization completed within 13 h, while in the case of ITC, the completion occurred in 3 h. This observation is in clear conflict with the authors' main claim in the manuscript's introduction that the crystallization time is decreased from days to hours using PPG.

Reply:

Thanks for the reviewer's comments. The comments of 10 and 3 by the reviewer are about the speed-up growth of the OIHPs single crystals. We would like to clarify it herein: This PC route is not to further increase the growth speed of ITC, but to produce high quality OIHP single crystals in a general approach and simultaneously maintain the high growth speed. At the meantime of high growth rate, PC route can effectively decrease the nucleation rate and reduce the defects and/or impurity phase resulted from the high growth speed. With the well-controlled nucleation process through PC, high quality OIHP single crystals can be achieved as demonstrated in **Fig. 1-2** in the main text by FAPbBr₃, FAPbI₃ and the mixed MA/FAPbBr₃ single crystals.

Comment 11:

The brand and purity of the PPG was not stated. Please state the brand, molecular weight, and purity of PPG.

Reply:

The detail experimental information is provided in **Table R6 (Supplementary Table 2)**. The glycol polymers are used as purchased without further treatment.

Table R6. (Supplementary Table 2). The information of the PPG, PEG, PAA, PVA and other additives

Type	Polymer/additive	Brand	Average Molecular Weight/Molecular Weight)	Purity
poly ethylene glycol	PEG-1500	Aladdin	1500	Analytical Purity
	PEG-2000	Aladdin	2000	Analytical Purity
	PEG-4000	Aladdin	4000	Analytical Purity
	PEG-6000	Aladdin	6000	Analytical Purity
poly propylene glycol	PPG-2000	Aladdin	2000	Analytical Purity
	PPG-3000	Aladdin	3000	Analytical Purity
	PPG-400	Macklin	400	Analytical Purity
polyacrylic acid	PAA-3000 (50% solution)	Aladdin	3000	Analytical Purity
polyvinyl alcohol	PVA-16000	Aladdin	16000	Analytical Purity
pentadecane	Pentadecane	Macklin	212	Analytical Purity
eicosane	Eicosane	Macklin	283	Analytical Purity

polystyrene	PS-16000	Aladdin	16000	Analytical Purity
-------------	----------	---------	-------	-------------------

Comment 12:

The description of the method for growing each type of crystal is cursory. It should be expanded.

Reply:

Thanks for the comments. The corresponding information is provided herein and in the Experimental Section in the revised manuscript.

Synthesis of perovskite powders FAPbX₃ (X = I, Br) powders were prepared by reacting formamidine acetate salt (FAAc), lead (II) acetate trihydrate (Pb(Ac)₂•3H₂O), hydriodic acid, or hydrobromic acid with the molar ratio about 1.1 : 1 : 6. A slight surplus of FAAc is to avoid the production of lead (II) iodide. Firstly, the Pb(Ac)₂•3H₂O was dissolved by HX (X = I, Br) solutions under stirring to obtain clear solution in a flask at 80 °C. Next, FAAc was added to the clear solution. Then black precipitate for FAPbI₃ or red precipitate for FAPbBr₃ was produced in the bottom after 1 - 2 h stirring and heating at 80 °C. Finally, the powders were collected using Büchner funnel filtration, washed by anhydrous ethanol for several times, and subsequently dried at 80 °C for 24 h. MAPbX₃ (X = I, Br, Cl) powders were prepared using the similar method as that of FAPbX₃. The difference is that in the synthesis process of MAPbX₃, the methylamine (CH₃NH₂) (40 wt.% in water) was used instead of the formamidine acetate salt. Excessive hydrochloric acid was used to make sure lead (II) acetate trihydrate completely dissolved. The MAPbCl₃ powder was obtained by cooling the solution for several days. CsPbBr₃ powder could also be prepared using cesium acetate and hydrobromic acid as an inorganic source and halogen source, respectively. The mixed-halide perovskite powders were prepared by mixing hydroiodic and hydrobromic acid or hydrobromic and hydrochloric acid as a mixed source of halogen. The mixed-organic cation perovskite powders were prepared by adding formamidine acetate salt and methylamine with different ratios. Finally, the mixed MAPbI_xBr_{3-x}, MAPbBr_xCl_{3-x} and MA_xFA_{1-x}PbX₃ (X = I, Br) perovskite powders were obtained.

The crystallization of perovskite single crystals. The obtained powders were dissolved in appropriate organic solvents with different concentrations for the growth of single crystals. FAPbI₃ and MAPbI₃ powders were dissolved in GBL for 0.75 g/mL, MAPbBr₃ powder was dissolved in DMF for 0.375 g/mL, FAPbBr₃ powder was dissolved in GBL/DMF (1:1) for 0.25 g/mL, MAPbCl₃ powder was dissolved in DMSO for 0.5 g/mL and CsPbBr₃ powder was dissolved in DMSO for 0.5 g/mL and 0.35 g/mL, respectively. After stirring for 3-4 hours for a complete dissolution, a certain amount (0.01-0.1 g/mL) of polymers was added. Finally, the precursor solution was filtered using a polytetrafluoroethylene filter with a 0.2 μm pore size (Whatman) and then placed on a hot plate preheated to a certain temperature for crystallization. FAPbI₃, MAPbI₃, MAPbBr₃ and FAPbBr₃ precursor solutions were treated at 60-95 °C for ITC crystallization. MAPbCl₃ and CsPbBr₃ precursor solutions were treated at room

temperature for evaporative crystallization. The PAA-3000 (50% solution) is dried in a vacuum drying oven and subsequently ground into powder before being added to the solution. Detailed information is shown in **Table R5 (Supplementary Table S3)**.

Table R5. (Supplementary Table S3). A more detailed description of the growth conditions for each type of OIHPs single crystals.

OIHPs	C _{OIHPs} (g/mL)	Solvent	Polymer	C _{Polymer} (g/mL)	Temperature (°C)
FAPbI ₃	0.75	GBL	PPG-3000	0.05	90
FAPbBr ₃	0.5	GBL/DMF (v:v = 1:1)	PEG-1500	0.05	60
MAPbI ₃	0.75	GBL	PPG-3000	0.03	95
MAPbBr ₃	0.6	DMF	PPG-3000	0.01	62
MAPbCl ₃	0.5	DMSO	PPG-3000	0.1	25
CsPbBr ₃	0.35	DMSO	PPG-3000	0.08	25
MAPbI _{2.1} Br _{0.9}	0.75	GBL/DMF (v:v = 1:1)	PPG-3000	0.02	90
MAPbI _{1.6} Br _{1.4}	0.75	GBL/DMF (v:v = 1:1)	PPG-3000	0.1	90
MAPbI _{0.12} Br _{2.88}	0.3	GBL/DMF (v:v = 1:1)	PPG-3000	0.1	90
MAPbBr _{2.85} Cl _{0.15}	0.55	DMF	PPG-3000	0.01	60
MAPbBr _{2.46} Cl _{0.54}	0.40	DMF	PPG-3000	0.01	60
MAPbBr _{2.1} Cl _{0.9}	0.37	DMF	PPG-3000	0.01	60
MA _{0.57} FA _{0.43} PbI ₃	0.75	GBL	PPG-3000	0.06	90
MA _{0.31} FA _{0.69} PbI ₃	0.75	GBL	PPG-3000	0.06	90
MA _{0.26} FA _{0.74} PbI ₃	0.75	GBL	PPG-3000	0.1	85
MA _{0.11} FA _{0.89} PbI ₃	0.75	GBL	PPG-3000	0.1	85

Reviewer #3 (Remarks to the Author):

Single crystals of organic-inorganic hybrid methylammonium lead trihalide perovskites show remarkably physical properties in photovoltaic applications such as high efficient photo-electric transition efficiencies, low trap density and charge transport properties. However, the growth of high-quality organic-inorganic halide perovskite semiconductor single crystals is a slow process, normally takes days or even weeks. The growth of distinctive mixtures on A or X sites for ABX₃ single crystals are also difficult, particularly, the growth of all family single crystals are impossible yet, though distinctive and various nucleation routes and growth mechanisms have been developed and proposed. The rapid growth of large size and high quality organic-inorganic perovskite single crystals are highly valuable, particularly when recent rapid developments of these materials for applications in x-ray and γ -ray detectors and sensors emerged.

The authors reported a novel polymer control nucleation route to develop an efficient approach for large-size and high-quality organic-inorganic perovskite single crystal growth. They synthesized high quality centimeter sized MA/FAPbBr₃ single crystals for the first time, and further, the whole-family high-quality single crystals including lead-based perovskite, mixed organic cations (FA/MA) and halide (I/Br, Br/Cl) single crystals. The polymer control nucleation mechanisms by coordinative interaction between the oxygen group of polymer polypropylene glycol with Pb²⁺ ions are also clarified by detail kinetic researches and in situ TEM and other systematic experimental observation. In the future, large scale growth of single crystals of all family MA/FAPbX₃ can be expected.

The results are novel, very high quality and extremely important. It is suggested to publish it after minor revision:

Many thank for the reviewer's very positive comments.

Comment 1:

In Figure 1 b, d, e and the related text, the x values in the MA_xFA_{1-x}PbX₃ (X = I, Br) should be denoted;

Reply:

Fig. 1 is revised. The MAPbI_xBr_{3-x} or MAPbBr_xCl_{3-x} single crystals were prepared and the I/Br and Br/Cl ratio were measured by X-ray Fluorescence (XRF). The MA_{1-x}FA_xPbI₃ or MA_{1-x}FA_xPbBr₃ single crystals were prepared and the MA/FA ratio were measured by Nuclear Magnetic Resonance (NMR).

Figure 1 | Photos of OIHPs single crystals. a, FAPbX₃ (X = I, Br), CsPbBr₃ and MAPbX₃ (X = I, Br, Cl) single crystals. b, A series of mixed halide MAPbI_xBr_{3-x} and MAPbBr_xCl_{3-x} single crystals. (0 ≤ x ≤ 3) c, d. Organic cations mixed MA_yFA_{1-y}PbX₃ (X = I, Br) single crystals. (0 ≤ y ≤ 1)

Comment 2:

The labels of x and y axes in the insets of Fig. 2, should be reformatted. The characters are too small to be noticed;

Reply:

Thanks for the referee's comments and suggestions. Fig. 2 is revised accordingly. More information is also included.

Figure 2 | *Carrier lifetime measurements.* Steady and transient state photoluminescence (PL) is measured using a 405 and 475 nm excitation wavelength, respectively. PL decay curves of a, the FAPbBr₃ single crystal and b, the FAPbI₃ single crystal. c, Comparison of carrier lifetimes of OIHPs single crystals. [26, 27, 31] d, The correlations between carrier lifetimes and average single crystal growth rates (in volume) from our work and literatures. [25, 26, 27, 28]

Comment 3:

In Page 8, equation 1 has no label yet and it should be (1); the others should be (2) and (3);

Reply:

The text is revised as below:

$$N = \frac{n}{V} = \left(\frac{\Delta M}{\bar{m}} \right) / V = \left(\frac{\Delta C \times V}{\frac{4}{3} \pi \bar{r}^3 \times \rho} \right) / V = \frac{\Delta C}{\frac{4}{3} \pi \bar{r}^3 \times \rho} \quad (1)$$

$$Y(t) = 1 - \exp \left[-(Kt)^n \right], K = K_0 \exp \left(-\frac{E_c}{RT} \right) \quad (2)$$

$$\ln(-\ln[1 - Y(t)]) = \ln K + n \ln t \quad (3)$$

The discussion of the classical Johnson-Mehl-Avrami-Kolmogorov (JMAK) model is removed because the results are not so important. Therefore, the equation 2, 3 are also removed.

Comment 4:

Comparing to other organic-inorganic perovskite single crystals, Why the growth of large size MA/FAPbBr₃ single crystals is difficult and was firstly reported in this work;

Reply:

Thanks for this critical point. The growth conditions of MAPbBr₃ and FAPbBr₃ reported in literature are different. This method cannot be directly extended to the MA/FA mixtures growth. In this study, we find a way to synthesize the single crystals towards a generalized method. Please also see Reply to Reviewer #2, Comment 1.

The defect structures of FA-based perovskite single crystals form easier than MA-based perovskite single crystals. So it is more challenging to grow large size FA-based single crystals. For example, only about 4 mm size FAPbBr₃ single crystals have been prepared [R9] while over 40 mm MAPbBr₃ single crystals can be obtained [R12]. Therefore, for the mixed MA/FAPbBr₃ single crystals with high FA contents, the induction period of nucleation was very short, for example, only 10 minutes, followed by forming large number of fine crystallites. Even if we lower the growth temperature and reduce the concentration, the obtained single crystal will be very faulty although the number of fine crystallites can be reduced.

So far, there are several approaches reported to grow MA/FAPbI₃ single crystals and 5 mm MA_{0.5}FA_{0.5}PbI₃ single crystals have been prepared. [R17] In our study, the PPG and PEG could greatly improve the stability of growth solution and avoiding the generation of defective structures. A series of MA/FAPbBr₃ single crystals are prepared with high quality (This reply file, Fig. R1) and nearly one-centimeter size MA_{0.86}FA_{0.14}PbBr₃ single crystal is prepared.

Comment 5:

The oxygen group of polymer polypropylene glycol plays a key role during the growth of organic-inorganic perovskite single crystals, is there other polymer with oxygen group may have the same behavior?

Reply:

The effect of glycol polymers is based on the interaction with the growth species., i.e., most important of Pb-O coordination bond. Therefore, other polymers with oxygen groups would also work with appropriate different conditions. As shown in Fig. R10, R11 (Supplementary Fig.S1, S2), polyethylene glycol (PEG), polyacrylic acid (PAA) or polyvinyl alcohol (PVA) also shows a similar effect and gives good result for the growth of some OIHPs single crystals.

Black phase α -FAPbI₃ crystals of 1-2 millimeters in size and ill-defined shapes could be obtained without adding polymers. As shown in Fig. R11, large size FAPbI₃ crystals with regular geometry shapes could be grown when adding an appropriate amounts and molecular weights of PEG, PPG, PVA, or PAA to perovskite precursor solutions.

However, other additives in comparison, such as pentadecane, eicosane, and polystyrene (PS), would not enhance the size or quality of perovskite crystals. Especially, the pentadecane could make the needle-like yellow crystals easy to grow.

Figure R10. (Supplementary Figure S1) Adding different polymers into FAPbI₃/GBL growth solution with different molecular weights and quantities. The diamonds represent positive effect on preparing big size FAPbI₃ crystals and the triangles represent no obvious positive effect. The blanks indicate the polymer could not reach this concentration in GBL. The empty slots in this illustration are due to the limit of solubility.

Figure R11. (Supplementary Figure S2) The photos of as-prepared FAPbI₃ single crystals when adding different polymers into the precursor solutions: (a) PEG-1500, (b) PPG-3000, (c) PAA-3000 and (d) PVA-16000.

Reviewer #4 (Remarks to the Author):

This work reports an interesting approach to controlling polymer nucleation process for highly efficient, large size and high-quality hybrid organic-inorganic perovskite single crystals. Compared with conventional methods based on solution temperature-lowering route, anti-solvent vapor assisted crystallization and slow evaporation, the method reported here present several notable advantages such as rapid crystal growth and readily applicable to crystals of varied compositions. Overall, this work is carried out with care. I would recommend its publication only if the authors could take into account the following points in revising the manuscript.

Many thanks for the reviewer's positive comments.

Comment 1:

In the introduction part, for the sake of a balanced overview, the authors should include the discussion of all-inorganic perovskite materials given their importance in various applications. For example, all-inorganic perovskite nanocrystals have found useful as X-ray scintillators (Nature 561, 88-93 (2018) doi:10.1038/s41586-018-0451-1).

Reply: Thanks for the comments. The reference is added as **ref. 18** in the main text.

Comment 2:

With the addition of PPG, the formation of the yellow phase of δ -FAPbI₃ was inhibited. What are the mechanisms during the growth for avoiding the δ -FAPbI₃? In addition, the author should provide evidence of the δ -FAPbI₃ without the PPG.

Reply:

We would discuss this issue based on the literatures and our experimental observation.

As Prof. Yangyang from UCLA reported, ^[R1] the crystallization of α -FAPbI₃ is an endothermic reaction and the appearance of light-yellow δ -phase FAPbI₃ crystals (**Fig. R12**) impeded the α -phase crystal growth. As shown in **Fig. 5**, the δ -phase FAPbI₃ is very easy to appear during the crystallization of α -FAPbI₃. Yangyang considered that fast crystal growth could lead to thermal fluctuation to form the light-yellow δ -phase FAPbI₃ and some needle-like NH₄PbI₃. ^[R1] The introduction of seed crystals is applied to reduce the thermal fluctuation and effectively hinder the growth of NH₄PbI₃ and δ -phase FAPbI₃. However, the introduced seed crystal could easily yield spontaneous nucleation.

In our work, the introduction of PPG could inhibit the appearance of δ -phase FAPbI₃ and other impurities. To study the thermal fluctuation during the nucleation process of α -FAPbI₃, the FAPbI₃/GBL, FAPbI₃/(GBL+PPG-3000) and FAPbI₃/(GBL+pentadecane) solutions are measured by DSC under an isothermal condition (90 °C). The results indicate that PPG greatly reduces the thermal fluctuation

during the nucleation process and finally decreased the nucleation rate.

Figure R12. (Supplementary Figure S3) (a) The powder XRD pattern of the needle-like yellow crystals from the FAPbI₃/GBL growth solution. (b) The experimental and calculated powder XRD patterns of δ -phase FAPbI₃ reported in literature. [R1]

Figure R13. The DSC scans showing endothermic peaks of different FAPbI₃ precursor solutions, the FAPbI₃/(GBL+PPG-3000) (blue line), the FAPbI₃/GBL (red line) and FAPbI₃/(GBL+pentadecane) (green), versus time recorded at 90 °C.

The DSC measurements are taken to observe the enthalpy change during the nucleation of FAPbI₃/GBL, FAPbI₃/(GBL+PPG-3000) and FAPbI₃/(GBL+pentadecane) precursor solutions at 90 °C. As shown in **Fig. R13**, at around 35 mins from the beginning, there is a sharp endothermic peak of FAPbI₃/GBL solution. A sharper endothermic peak of FAPbI₃/(GBL+pentadecane) is also observed at around 46 mins. During the crystallization of FAPbI₃/GBL and FAPbI₃/(GBL+pentadecane) precursor solutions, the first crystal also appears around this time frame. Therefore, we consider the endothermic peak belongs to the nucleation process. The enthalpy change (ΔH) of FAPbI₃/GBL solution is about 0.09 J/g which is less than the $\Delta H = 0.14$ J/g of FAPbI₃/(GBL+pentadecane) solution. Because the adding of pentadecane could make the δ -FAPbI₃ crystals easier to form during the crystallization. (**Fig. R13**) However,

there is no obvious endothermic peak of FAPbI₃/(GBL+PPG-3000) solution because of the low nucleation rate. The DSC experimental results are consistent with our conclusions that PPG-3000 obviously decreases the nucleation rate. The pentadecane increases the nucleation rate which promotes the formation of the yellow phase.

Comment 3:

In Fig. 3d, the authors believed that the variation in the intensities of the absorption is ascribed to the free iodine ions and the increased concentration of I³⁻, derived from the iodine ions of iodoplumbates that are replaced by oxygen groups of PPG. Was the oxygen of PPG coordinated to the octahedron Pb (II) ion? I would suggest additional characterizations for the sample, such as NMR or FTIR.

Reply:

This question is the same to the previous Comment 3 raised by Reviewer #1. Please see the reply to Comment 3 raised by Reviewer #1 for detailed information. We demonstrated the coordination of glycol polymers with Pb²⁺ by Raman and FTIR.

When the polymer is added into the solution, the coordination of the glycol polymers with Pb²⁺ forms and the absorption peak of PbI_n⁽ⁿ⁻²⁾⁻ become weak. (Fig. 3d in Main text) Therefore, it is reasonable to believe that the polymer replaced a partial of the iodine ions from the coordination of Pb²⁺. The iodine ions may be oxidized by the air to form some I³⁻ ions, therefore the increase of free iodine ions can be linked to the I³⁻ ions if other conditions do not change.

Comment 4:

An in-situ electrochemical impedance spectroscopy (EIS) method was applied to probe the kinetics of the solution's concentration. How was characterization performed (such as 0.02g/ml and 0.11g/ml). The details should be provided.

Reply:

The details are added to the supplementary information.

Two platinum electrodes were used as test electrodes and were inserted into the precursor organic solution for the EIS testing. The solutions were tested at a scanning frequency range of 2 MHz to 100 Hz. Based on the equivalent circuit model, the R_s (ohmic resistance of solution) can be calculated, which corresponds to the solution impedance moduli at high frequency end (|Z|). The results show that the R_s (ohmic resistance) represents the solution resistance based on the equivalent circuit model. The solution resistance is in accord with the ionic concentrations in a similar solution which could be used as a parameter of solute consumption at a fixed temperature. The EIS measurements were performed every ten minutes during the crystallization process at 90 °C.

Further analysis reveals that the R_s can be obtained using the high frequency resistance. The resistances of growth solutions with different concentration at 90 °C were tested using EIS at 10000 Hz. The fixed frequency measurements have the potential in real-time monitoring for being much faster with a full EIS scan. The solution concentration as a function of solution impedance moduli were obtained. Using the solution resistance obtained from the EIS profiles during crystallization, the real-time concentrations (C_R) of the growth solution could be calculated. As the original concentration of FAPbI_3 (C) is 0.75 g/mL, the solute consumption (C_0) could be calculated as following equation.

$$C_0 = C - C_R$$

Detailed experiment results are shown below:

- (1) the Nyquist plots of growth solutions during crystallization process. Based on the equivalent circuit model, the R_s can be calculated. (Fig. R14)
- (2) The solution resistance (R_s) as a function of growth time during the crystallization (Fig. R15).
- (3) The different concentrations of growth solution as a function of R_s estimated at high frequency impedance moduli (10000 Hz) (Fig. R16). Then, the C_R could be calculated combined with (2) and (3).
- (4) The solute consumption of $\text{FAPbI}_3/\text{GBL}$ solution (C_0) with or without PPG-3000 as a function of growth time at 90 °C are shown in Fig. R17.

Figure R14. (Supplementary Figure S17) Nyquist plots of FAPbI_3 GBL solutions (0.75 g/mL) with (a) or without PPG-3000 (b) (0.05 g/mL) at 90 °C. Inset is the equivalent circuit model.

Figure R15. (Supplementary Figure S16). Solution resistance (R_s) as a function of growth time during the crystallization of FAPbI₃ single crystals: (a) FAPbI₃/(GBL+PPG-3000) solution ($C_{\text{FAPbI}_3} = 0.75$ g/ mL, $C_{\text{PPG-3000}} = 0.05$ g/mL). (b) FAPbI₃/GBL solution ($C_{\text{FAPbI}_3} = 0.75$ g/ mL). (c) FAPbI₃+Pentadecane/GBL solution ($C_{\text{FAPbI}_3} = 0.75$ g/ mL, $C_{\text{Pentadecane}} = 0.05$ g/mL). Insets are the photos of crystals in growth solutions after the crystallization process.

Figure R16 (Supplementary Figure 18). The concentrations of FAPbI₃ in FAPbI₃/GBL(a) and FAPbI₃/(GBL+PPG-3000) (b) as a function of R_s estimated with high frequency impedance moduli at 10000 Hz. This relation is also used as a working curve to calculate the consumption of solutes.

Figure R17. (Fig. 5 in the main text) The crystallization kinetics analysis of FAPbI₃ single crystals by EIS. The solute consumption as a function of growth time at 90 °C. The FAPbI₃/GBL solution ($C_{\text{FAPbI}_3} = 0.75 \text{ g/mL}$): (1) only black crystals grow (red) or (2) yellow crystals also appear (green); The FAPbI₃/(GBL+PPG-3000) solution ($C_{\text{FAPbI}_3} = 0.75 \text{ g/mL}$, $C_{\text{PPG-3000}} = 0.05 \text{ g/mL}$) that only black crystal grows (blue). The data is calculated based on the EIS measurements. The yellow needle-like crystals, marked by the white arrow, is δ -FAPbI₃.

Comment 5:

In Fig. 5a, without the addition of the PPG, α -FAPbI₃ and δ -FAPbI₃ are likely to form. How do the authors distinguish one another and calculate the concentration?

Reply:

We can distinguish the crystals of α -FAPbI₃ and δ -FAPbI₃ with our eyes. (1) The colors of two phases are different. As shown in Fig. R18, the α -FAPbI₃ crystal is black while the δ -FAPbI₃ crystal is yellow. According to the literature, the black single crystal (α -FAPbI₃) is a cubic phase ($Pm\bar{3}m$ space group) with $a = 6.3620(8) \text{ \AA}$. [R18] The yellow crystal shows the hexagonal system ($P6_3mc$ space group) with $a = b = 0.866 \text{ nm}$ and $c = 0.790 \text{ nm}$. [R1]

We can also distinguish them by selected area electron diffraction (SAED). Furthermore, the nanoscaled α -FAPbI₃ and δ -FAPbI₃ in the growth solutions are also investigated by TEM as shown in Fig. R19, the α -FAPbI₃ nanoparticles (a) usually have a regular shape. However, the δ -FAPbI₃ nanoparticles are usually polycrystalline at this early stage.

Figure R18. The photos of (a) α -FAPbI₃ and (b) δ -FAPbI₃ crystals.

Figure R19. The TEM images and SAED of (a) α -FAPbI₃ and (b) δ -FAPbI₃ nanocrystals.

Comment 6:

What is the criterion for identifying point A and B? It is rather qualitative by choosing the appearance of the visible crystal.

Reply:

Thank you for this critical question. As shown in Fig. R20, the two points of A and B describe the crystallization processes based on the curve's shape. The differential plots show the relation of consumption rate versus time. The point that the differential curve with a minimum value and starts to go up is defined as point A, which indicates that a remarkable change in the solution results in a turning consumption rate. The cross point that the differential curve goes down and flats assigned by using the tangent lines is defined as point B, which means the final slow down for the growth process. It should be noted that the first appearance of a visible crystal by naked eyes is around the assigned the point A, showing the close relation of a growing single crystal with the acceleration in solute consumption. As for point B, the consumption of solute slows

down obviously because of the depletion in supersaturation, and since the regular crystal growth comes to a stop, it marks a good timing to harvest the product before unwanted second growth.

Figure R20 (Supplementary Figure S19). The solute consumption as a function of growth time at 90 °C and the differential curves of the solute consumption. The $\text{FAPbI}_3/\text{GBL}$ solution ($C_{\text{FAPbI}_3} = 0.75 \text{ g/mL}$): (1) only black crystals grow (red) or (2) yellow crystals also appear (green); The $\text{FAPbI}_3/(\text{GBL}+\text{PPG-3000})$ solution ($C_{\text{FAPbI}_3} = 0.75 \text{ g/mL}$, $C_{\text{PPG-3000}} = 0.05 \text{ g/mL}$) that only black crystal grows (blue).

References

- [R1] Han, Q. F. *et al.* Single Crystal Formamidinium Lead Iodide (FAPbI₃): Insight into the Structural, Optical, and Electrical Properties. *Adv. Mater.* **28**, 2253-2258 (2016).
- [R2] Yang, M. M. *et al.* A Raman spectroscopic study of lead and zinc acetate complexes in hydrothermal solutions. *Geochim. Cosmochim. Acta.* **53**, 319-326 (1989).
- [R3] Rozenberg, M., Loewenschuss, A., Marcus, Y. IR spectra and hydration of short-chain polyethyleneglycols. *Spectrochim. Acta Part A.* **54**, 1819-1826 (1998).
- [R4] Shameli, K. *et al.* Synthesis and Characterization of Polyethylene Glycol Mediated Silver Nanoparticles by the Green Method. *Inter. J. Molecul. Sci.* **13**, 6639-6650 (2012).
- [R5] Maculan, G. *et al.* CH₃NH₃PbCl₃ single crystals: inverse temperature crystallization and visible-blind UV-photodetector. *J. Phys. Chem. Lett.* **6**, 3781-3786 (2015).
- [R6] Wei, H. *et al.* Sensitive X-ray detectors made of methylammonium lead tribromide perovskite single crystals. *Nat. Photon.* **10**, 333-339 (2016).
- [R7] Yakunin, S. *et al.* Detection of gamma photons using solution-grown single crystals of hybrid lead halide perovskites. *Nat. Photon.* **10**, 585-589 (2016).
- [R8] Johns, P. M. *et al.* Enhanced gamma ray sensitivity in bismuth triiodide sensors through volumetric defect control. *Appl. Phys. Lett.* **109**, 092105 (2016).
- [R9] Zhumekenov, A. *et al.* Formamidinium lead halide perovskite crystals with unprecedented long carrier dynamics and diffusion length. *ACS Energy Lett.* **1**, 32-37 (2016).
- [R10] Huang, Y. *et al.* The Intrinsic Properties of FA_{1-x}MA_xPbI₃ Perovskite Single Crystals. *J. Mater. Chem. A.* **5**, 8537-8544 (2017).
- [R11] Saidaminov, M. I. *et al.* High-quality bulk hybrid perovskite single crystals within minutes by inverse temperature crystallization. *Nat. Commun.* **6**, No. 7586 (2015).
- [R12] Liu, Y. C. *et al.* Low-temperature-gradient crystallization for multi-inch high-quality perovskite single crystals for record performance photodetectors. *Mater. Today*, **22**, 67-75 (2019).
- [R13] Saidaminov, M. I. *et al.* Inorganic lead halide perovskite single crystals: phase-selective low-temperature growth, carrier transport properties, and self-powered photodetection. *Adv. Optical. Mater.* 1600704. (2017).
- [R14] Dong, Q. F. *et al.* Electron-hole diffusion lengths > 175 μm in solution-grown CH₃NH₃PbI₃ single crystals. *Science* **347**, 967-970 (2015).
- [R15] Shi, D. *et al.* Low trap-state density and long carrier diffusion in organolead trihalide perovskite single crystals. *Science* **347**, 519-522 (2015).
- [R16] Yao, F. *et al.* Room-temperature liquid diffused separation induced crystallization for high-quality perovskite single crystals. *Nat. Commun.* **11**, 1194 (2020).
- [R17] Li, W. G. *et al.* A formamidinium-methylammonium lead iodide perovskite single crystal exhibiting exceptional optoelectronic properties and long-term stability. *Journal of Materials Chemistry A*, **5**, 19431 (2017).
- [R18] Weller, M. T. *et al.* Cubic Perovskite Structure of Black Formamidinium Lead Iodide, α-[HC(NH₂)₂]PbI₃, at 298 K. *J. Phys. Chem. Lett.* **6**, 3209-3212 (2015).

REVIEWERS' COMMENTS

Reviewer #1 (Remarks to the Author):

In this revised manuscript, the authors have specifically addressed all reviewers' concerns. I am fully satisfied with the answers. I recommend accept it for publication without further changes.

Reviewer #2 (Remarks to the Author):

Although this work is carried out with care, the authors' report of growing large perovskite single crystals with improved photophysical properties is an incremental improvement over the currently available methods for growing large perovskite single crystals. The work is more suitable for a more specialized journal.

Reviewer #3 (Remarks to the Author):

The authors have revised the manuscript according to the Reviewers suggestions, and the paper can be accepted for publication now.

Reviewer #4 (Remarks to the Author):

Authors have addressed all my concerns in this revised manuscript. I recommend its publication.